# CIAR: Interval-based Collaborative Decoding for Image Generation Acceleration

**Keming Ye[1], Zhou Zhao[1], Fan Wu[2], Shengyu Zhang[1]**

[1]Zhejiang University
[2]Shanghai Jiao Tong University

{kemingye, zhaozhou, sy_zhang}@zju.edu.cn
wu-fan@sjtu.edu.cn

## Abstract

Auto-regressive (AR) models have recently made notable progress in image generation, achieving performance comparable to diffusion-based approaches. However, their computational intensity and sequential nature impede on-device deployment, causing disruptive latency. We address this via a cloud-device collaboration framework **CIAR**, which utilizes on-device self-verification to handle two key properties of visual synthesis: *the vast token vocabulary* required for high-fidelity images and *inherent spatial redundancy* which leads to extreme predictability in homogeneous regions, while object boundaries exhibit high uncertainty. Uniform verification wastes resources on such redundant tokens. Our solution centers on an on-device token uncertainty quantifier, which adopts continuous probability intervals to accelerate processing and make it feasible for large visual vocabularies instead of conventional discrete solution sets. Additionally, we incorporate a Interval-enhanced decoding module to further speed up decoding while maintaining visual fidelity and semantic consistency via a distribution alignment training strategy. Extensive experiments demonstrate that CIAR achieves a 2.18× speedup and reduces cloud requests by 70%, while preserving image quality compared to existing methods.

## 1 Introduction

Auto-regressive (AR) models have emerged as a powerful paradigm for high-fidelity image synthesis (Brown et al., 2020; Touvron et al., 2023), with architectures like Anole and LlamaGen (Sun et al., 2024; Chern et al., 2024) achieving remarkable results rivaling diffusion models (Ho et al., 2020; Rombach et al., 2022). These approaches leverage discrete token representations that capture rich visual semantics through codebooks of increasing size, enabling unprecedented photorealism. However, despite their capabilities, AR models suffer from critical limitations that impede their practical deployment. (Van Den Oord et al., 2017; Esser et al., 2021) A key challenge lies in their substantial parameter size, driven by the expanding codebooks used for tokenization, which makes it hard for on-device deployment. Furthermore, the inherent sequential nature of AR models, requiring one-by-one token generation, results in slow inference speeds that degrade the users' experience, especially on resource-constrained devices.

A promising direction to address these practical limitations is to leverage a cloud-device collaborative framework. In this setup, a lightweight AR model is deployed on the device for fast token generation, while a larger model on the cloud performs token verification in parallel, similar to the speculative decoding (SD) paradigm (Chen et al., 2023; Cai et al., 2023). Several developments in visual speculative decoding provide a feasible foundation for this architecture, offering the potential to combine the efficiency of device computing with the power of cloud-based models for real-time image generation. (Wang et al., 2024b; Jang et al., 2024)

However, two fundamental challenges critically undermine conventional cloud-device collaboration for AR image generation. The first is the *prohibitive overhead from high-volume token verifica-*

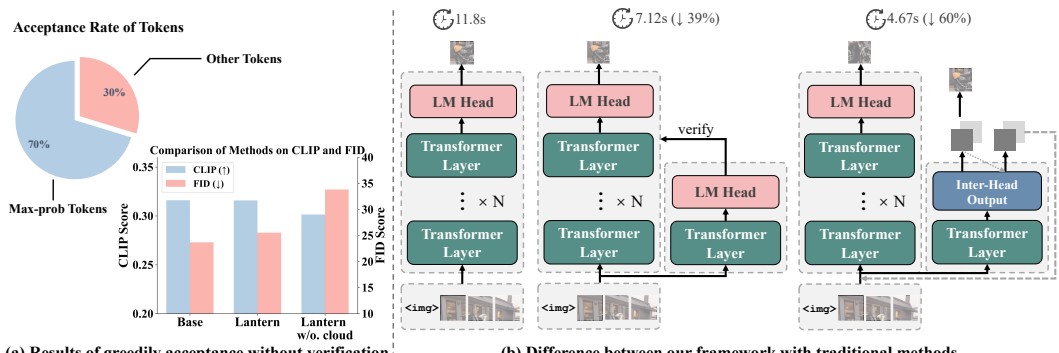

Figure 1: (a) Acceptance analysis of Lantern. The pie chart shows the ratio of max-prob vs. other tokens, and the bar chart compares Lantern without verification to the baseline. (b) Comparison of decoding frameworks. From left to right: baseline, Lantern, and our CIAR with Inter-Head and cloud-device collaboration, which reduces latency while preserving output quality.

*tion.* Unlike text, image tokens scale quadratically with resolution, creating a data explosion that establishes the network as the primary bottleneck (Van Den Oord et al., 2016; Chang et al., 2022). This not only negates the cloud's computational advantage but also incurs substantial energy and operational costs, rendering the framework impractical for real-world use. The second is the *inefficiency of the uniform verification policy.* Conventional verification employs an indiscriminate, token-by-token validation process (Zhang et al., 2025), treating every part of the image as equally important. This approach is misaligned with the intrinsic properties of visual data, which exhibit spatially heterogeneous uncertainty (Kendall & Gal, 2017). Vast regions of an image, such as backgrounds and smooth surfaces, contain highly redundant and predictable information (low-entropy regions). Verifying tokens in these areas squanders communication cost and cloud resources on overwhelmingly correct predictions (Bolya et al., 2022). Conversely, semantically critical areas like object boundaries and complex textures introduce high localized uncertainty where generation errors are most likely to occur and propagate. The uniform verification strategy fails to distinguish between these cases, resulting in a suboptimal trade-off: it wastefully validates the predictable while failing to concentrate scrutiny on the uncertain, thereby limiting both speed-up potential and quality assurance.

Building on our prior analysis, we first explore the limits of pure on-device decoding. As shown in Figure 1(a), when the device model greedily selects the top-probability token, overall scene semantics are preserved but perceptual quality degrades markedly, with visible artifacts and blurring in fine details (Dahl et al., 2017). A closer inspection reveals that approximately 70% tokens approved by the cloud already match the device's top-output, indicating that most tokens can be decided locally, while a subset of detail-related tokens still benefits from cloud correction (Liu et al., 2023; Zhang et al., 2023). Motivated by this finding, we hope to insert a lightweight uncertainty quantifier on the device for self-verification. However, off-the-shelf entropy measuresAli et al. (2025); Nikitin et al. (2024); Stolte et al. (2025) are ineffective over large visual vocabularies and ignore spatial context, thereby masking true ambiguity. Moreover, once the device commits to a verified prefix, its conditional distribution may drift from the cloud model's expectations, impairing subsequent decoding (Bengio et al., 2015). Thus, it is essential to align the device and cloud distributions.

To tackle these issues, we introduce **CIAR**—a **C**ollaborative **I**nterval-based **A**uto**R**egressive Decoding framework designed for efficient, high-fidelity visual synthesis. Our framework consists of two modules: on-device Interval-based uncertainty quantifier and Interval-enhanced decoding module, and introduces a distribution alignment training strategy to support them.

**Interval-Based uncertainty quantifier** measures token uncertainty on device, capturing continuous visual distribution characteristics while improving computational efficiency. At its core is our *Inter-Head*, which outputs upper and lower probability intervals for each token. By using continuous probability intervals instead of discrete solutions, it preserves the continuous nature of visual tokens for more precise uncertainty quantification and significantly reduces time complexity. **Cloud-Enhanced decoding module** facilitates collaboration between cloud and device to maintain

distribution alignment. Excessive local token generation can lead to distribution drift, impairing visual coherence. We counteract this by incorporating prefix injection and interval-based features for *enhanced-decoding*, which jointly constrain distribution shift and sustain high image quality. **Distribution alignment training strategy** is designed to train the Inter-Head for alignment with the cloud distribution. Since the Inter-Head outputs probability intervals,which is distinct from the cloud module, we introduce an *interval-aware DRO* variant, employing an alignment loss computed over both lower and upper bound logits, progressively reducing divergence between distributions and ensuring output consistency.

Our contributions can be summarized as follows:

1. We propose **CIAR**, a collaborative interval-based autoregressive decoding framework that leverages on-device uncertainty-aware verification to eliminate redundant uniform verification and accelerate image synthesis while maintaining high fidelity.

2. We design an **Inter-Head** for continuous interval estimation, which derives upper and lower probability intervals from logits, and an **Cloud-Enhanced decoding** module that uses collaborative feature guidance for enhanced-decoding, supported by an interval-aware DRO training strategy, aligning cloud and device distribution despite structural discrepancy.

3. We conduct extensive experiments, demonstrating that CIAR achieves a **2.18× speed-up** and reduces cloud requests by **70%** compared to state-of-the-art speculative decoding methods, without compromising visual fidelity.

## 2 RELATED WORKS

**Visual Auto-Regressive Models.** Visual AR models extend language model paradigms to image generation by converting images into discrete token sequences (Van Den Oord et al., 2017; Esser et al., 2021). Early works with raster scan orders suffered high cost and inferior quality compared to diffusion models. More recent methods such as VQVAE/VQGAN-based AR models like LlamaGen (Sun et al., 2024) utilize large vocabularies and improved tokenizers to produce high-fidelity next-token predictions. Models like Anole (Chern et al., 2024) further support native multimodal and interleaved image-text generation with fine-tuned visual tokens. Despite the advances, these models remain too slow and resource-intensive for on-device use under large vocabulary, long sequence settings.

**Cloud-Device Collaborative Inference.** Cloud-device collaboration balances computation between powerful cloud servers and resource-limited devices. Federated Learning (McMahan et al., 2017) is a common approach, enabling local training with cloud-based gradient aggregation while preserving privacy, but it struggles with real-time distribution shifts. Other works focus on device adaptation: DCCL (Yao et al., 2021) performs cloud pre-training followed by on-device fine-tuning, while DUET (Lv et al., 2023b) uses a cloud hypernetwork to generate personalized weights from device data. More interactive paradigms have also emerged, such as CD-CCA (Wang et al., 2024a), which uploads high-uncertainty samples for cloud analysis, and DC-CCL (Ding et al., 2023), which splits large models into cloud and device modules and refines both via knowledge distillation.

## 3 METHODOLOGY

In this section, we introduce CIAR, a collaborative interval-based autoregressive decoding framework, designed to accelerate on-device decoding while enhancing visual synthesis quality. We first define the problem setup, then describe the overall framework and detail the key components.

### 3.1 PRELIMINARY

We denote the cloud-side AR model by *Cloud AR* with conditional distribution $q_{\text{cloud}}(x_t|x_{<t}, T)$, and the device model by *Device AR* with $q_{\text{device}}(x_t|x_{<t}, T)$. Here $T = (t_1, \ldots, t_L)$ is the tokenized text prompt and $X = (x_1, \ldots, x_K)$ is the sequence of image tokens. Each token $x_t$ takes values in $\mathcal{C} = \{c_1, \ldots, c_L\}$, where $L$ is the size of the visual codebook. In text-to-image generation, the

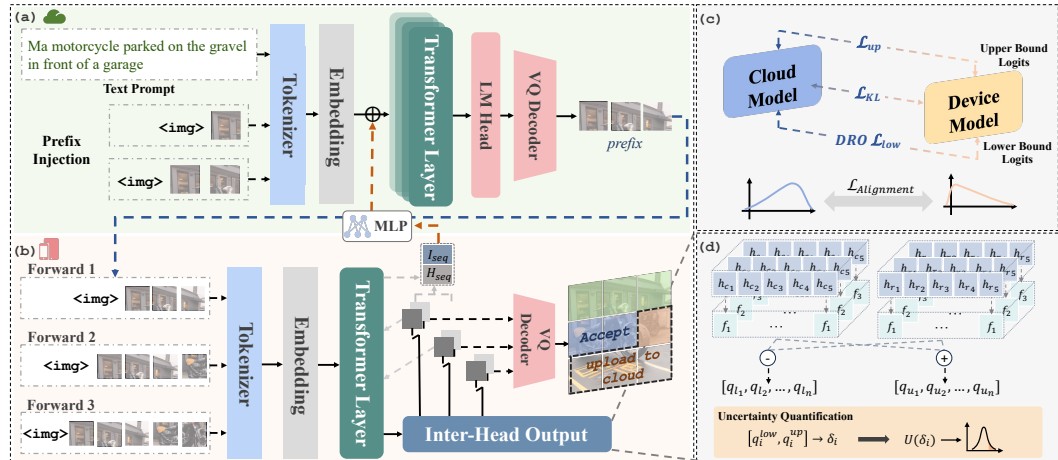

Figure 2: Overview of CIAR. (a) The cloud-side AR model generates image token prefixes from the input prompt. These prefixes are then sent to (b) a lightweight device model with Inter-Head accepts confident tokens locally and sends uncertain ones with interval features to the cloud for verification and distribution alignment. (c) Interval-Based Alignment Strategy. (d) Computation of uncertainty intervals in the Inter-Head.

model samples

$$q(X \mid T) = \prod_{t=1}^{K} p(x_t \mid x_{<t}, T), \text{where} \quad K = h \times w, \quad h = \frac{H}{f}, \quad w = \frac{W}{f} \tag{1}$$

for an $H \times W$ image and downsampling factor $f$. Prediction is strictly autoregressive: each $x_t$ depends only on its predecessors and the prompt. Once $(x_1, \ldots, x_K)$ are generated, they are mapped to embeddings via a learned codebook:

$$E = \{ e_i \in \mathbb{R}^D : i = 1, \ldots, V \}, \tag{2}$$

so that $x_t \rightarrow e_{x_t}$. The embeddings are then reshaped into a tensor of shape $(h, w, D)$ in raster-scan order and passed to a decoder (e.g., VQVAE or VQGAN) to reconstruct the final image.

## 3.2 CIAR: COLLABORATIVE INTERVAL-BASED AUTOREGRESSIVE DECODING

CIAR consists of three key components: an on-device *Interval-Based uncertainty quantifier* powered by a lightweight *Inter-Head*; an *Cloud-Enhanced decoding module* on the cloud that refines generation using device-provided interval cues; and a *distribution alignment training strategy* that aligns token distributions between device and cloud.

### 3.2.1 DEVICE-SIDE UNCERTAINTY QUANTIFICATION

The device uses a lightweight AR model for fast token generation. To identify reliable tokens, precise uncertainty quantification is essential. Traditional entropy-based methods are ill-suited for visual synthesis due to two inherent visual properties: (1) large codebooks yield flat probability distributions, and (2) spatial continuity leads to similar entropy among related tokens. We thus propose the **Interval Head (Inter-Head)** to estimate uncertainty via continuous probability intervals.

**Interval Head Architecture.** Different from standard LM Head, Inter-Head extends the output dimension to $2 \times |\mathcal{V}|$ where $\mathcal{V}$ is the vocabulary. Formally, given hidden state $\mathbf{h}_t$ for token $t$:

$$\mathbf{c}_t = \text{Linear}_{\text{center}}(\mathbf{h}_t), \quad \mathbf{r}_t = \text{Softplus}\left(\text{Linear}_{\text{radius}}(\mathbf{h}_t)\right), \quad \mathbf{p}_t^I = \textbf{InterFuse}(\mathbf{c}_t, \mathbf{r}_t). \tag{3}$$

The use of a Softplus function ensures the radius $r_t$ is strictly non-negative. This defines a logit interval $[l_t^l, l_t^u] = [c_t - r_t, c_t + r_t]$. We define a new operator **InterFuse** to yield the final probability interval $\mathcal{P}_t = [p_t^l, p_t^u]$, which is guaranteed to satisfy the properties of a valid solution set, namely $\sum_i p_i^l \leq 1 \leq \sum_i p_i^u$. A detailed proof is provided in the Appendix A.4.

**Uncertainty Quantification.** Given the probability interval $\mathcal{P}_t$, we require a metric that converts it into a scalar measure of uncertainty. A simple average is inadequate, as token uncertainty depends on both the total volume and the dispersion of the interval widths. Based on properties observed in visual autoregressive generation, we find that uncertainty increases with a larger total interval volume and rises with greater variance among interval widths. A detailed analysis of these principles is provided in the Appendix A.3.

Adhering to these principles, we formulate a novel uncertainty score. Let $\delta_t = p_t^u - p_t^l \in \mathbb{R}_{\geq 0}^{|\mathcal{V}|}$ be the vector of probability interval widths. We first define the *Total Ambiguity Volume*, $\Omega_t$, as the $L_1$ norm of the widths, and the *Confidence Disparity*, $\Sigma_t$, as the standard deviation of the widths. The final uncertainty score $\mathcal{U}(\mathcal{P}_t)$ is their product:

$$\mathcal{U}(\mathcal{P}_t) = \underbrace{\|\delta_t\|_1}_{\Omega_t : \text{Total Ambiguity}} \cdot \underbrace{\sqrt{\frac{1}{|\mathcal{V}|} \sum_{i=1}^{|\mathcal{V}|} \left(\delta_{t,i} - \bar{\delta}_t\right)^2}}_{\Sigma_t : \text{Confidence Disparity}} \tag{4}$$

where $\bar{\delta}_t$ is the mean of the interval widths $\delta_{t,i}$. This formulation ensures that the score is high only when both the total ambiguity and the disparity are significant, providing a robust and sensitive measure of uncertainty. This score is then used by a dynamic thresholding policy to make the final determination for token acceptance.

### 3.2.2 CLOUD-ENHANCED DECODING

Early in generation, limited context on the device often causes deviation from the target distribution, while later, substantial local tokens may lead the cloud predictions to drift. To mitigate this, we introduce a *Cloud-Enhanced Decoding module* combining Prefix Injection and Interval-enhanced Decoding.

**Prefix Injection.** To improve initial generation quality and reduce unnecessary cloud requests, the cloud pre-generates a short prefix $\{y_1, \ldots, y_m\}$ where $m = \lfloor \rho \cdot T \rfloor$, with $\rho$ as the prefix rate. This prefix provides high-quality contextual guidance to the device, constraining local generation and alleviating distribution drift. The prefix length $\rho$ involves a trade-off: larger $\rho$ improves alignment and fidelity but increases cloud latency, while smaller $\rho$ speeds up decoding at the cost of weaker guidance. Empirically, a moderate $\rho$ achieves the best balance, significantly lowering cloud queries while maintaining visual quality.

**Interval-enhanced Decoding.** We introduce *Interval Feature Injection* to incorporate device-side uncertainty into cloud verification. For each locally accepted token $x_{t+i}$, the device's Inter-Head produces center and radius logits $(c_{t+i}, r_{t+i})$, defining a probability interval $\mathcal{P}_{t+i}$. A lightweight projection network $\phi$ maps the token's hidden state $f_{t+i}$ and its interval $\mathcal{P}_{t+i}$ to a compact **Interval Feature** $f_{t+i}^I$:

$$f_{t+i}^I = \phi(\mathbf{Concat}\,(f_{t+i}, \mathcal{P}_{t+i})) \in \mathbb{R}^d. \tag{5}$$

This interval feature serves as structured, actionable intelligence about the device's confidence.

During the verification or resampling phase on the cloud, this auxiliary feature is injected into the cloud model's decoder. As the cloud model processes the token $x_{t+i}$, its decoder's computation is conditioned not only on the token's embedding $E(x_{t+i})$ but also on our interval feature. The update rule for the cloud's hidden state $h_{t+i+1}^C$ is thus modified:

$$h_{t+i+1}^C = \text{Decoder}^C \left(E(x_{t+i}) + f_{t+i}^I\right) \tag{6}$$

By augmenting the input with this confidence-aware feature, we actively steer the cloud's generative process. This maintains distributional alignment and prevents error accumulation, enhancing output consistency and visual continuity.

### 3.2.3 INTERVAL-BASED ALIGNMENT STRATEGY

Since the Inter-Head architecture differs from the cloud's output layer, parameter sharing is infeasible. This necessitates a training strategy that simultaneously: (1) ensures accurate probability

interval estimation, and (2) maintains distributional alignment with the cloud model. The nature of this task, which is to simultaneously optimize for an optimistic (upper bound) and a pessimistic (lower bound) outcome, is conceptually analogous to finding a robust solution that performs well even in a worst-case scenario. This parallel naturally draws inspiration from the principles of Distributionally Robust Optimization (DRO). Therefore, we propose the **Inter-DRO Loss** which can operate on probability intervals, to address this alignment challenge.

**Classical Distributionally Robust Optimization.** Traditional DRO improves model robustness by optimizing against distribution shifts. Instead of minimizing loss under a single distribution, DRO minimizes the worst-case expected loss over an uncertainty set $\mathcal{U}$ of plausible distributions. Formally, the objective for a model with parameters $\theta$ is to solve the minimax problem:

$$\min_\theta \left\{ \sup_{P \in \mathcal{U}} \mathbb{E}_{z \sim P}[\mathcal{L}(f_\theta(z))] \right\} \tag{7}$$

where $f_\theta$ is the model, $z$ represents a data sample, and $\mathcal{L}$ is the loss function.

In practice, this abstract formulation is often operationalized through more concrete settings. One prominent example is Group DRO, where the worst-case distribution is selected from a finite set of known data groups $\mathcal{G} = \{1, \dots, m\}$. The objective then simplifies to minimizing the maximum expected loss across all groups:

$$\min_\theta \max_{g \in \mathcal{G}} \mathbb{E}_{z \sim P_g}[\mathcal{L}(f_\theta(z))] \tag{8}$$

This core principle of optimizing against a worst-case scenario forms the conceptual foundation for our Inter-DRO loss, which we will adapt to the unique structure of our probability intervals.

**Inter-DRO Loss** Our training objective for the **Interval Head** operationalizes Distributionally Robust Optimization (DRO) principles by treating the predicted lower probability bound $q^L$ as the worst-case scenario. This interpretation naturally aligns with DRO's minimax philosophy. Unlike standard language modeling heads, the Interval Head outputs a valid probability interval parameterized by center $\mathbf{c}$ and radius $\mathbf{r}$ logits, requiring a multi-faceted loss design.

To ensure predictive accuracy and cloud-device alignment, we minimize divergence from cloud outputs $\mathbf{o}_{\text{cloud}}$ via anchor loss:

$$\mathcal{L}_{\text{anchor}} = \lambda_v \|p - p_{\text{cloud}}\|_1 + \lambda_p \text{CE}(p_{\text{cloud}}, p) \tag{9}$$

where **CE** denotes cross-entropy, $p = \text{softmax}(\mathbf{o})$, and $\mathbf{o} \in \{\mathbf{o}^{\text{up}}, \mathbf{o}^{\text{lo}}\}$ for respective bounds.

For the lower bound, which represents the worst-case prediction, we enhance the anchoring loss with a DRO strategy. We employ an adversarial reweighting scheme where samples in the batch that are harder to predict (i.e., have a higher loss) are given greater weight:

$$\mathcal{L}_{\text{lo}}^{\text{DRO}} = \max_{\boldsymbol{w} \in \mathbb{S}} \sum_{n=1}^{N} w_n \text{CE}(p_{\text{cloud}}^{(n)}, p_{\text{lo}}^{(n)}), \quad w_n = \frac{\exp\left(\alpha \text{CE}(p_{\text{cloud}}^{(n)}, p_{\text{lo}}^{(n)})\right)}{\sum_{k=1}^{N} \exp\left(\alpha \text{CE}(p_{\text{cloud}}^{(k)}, p_{\text{lo}}^{(k)})\right)} \tag{10}$$

where $w_n$ is the weight derived from per-sample risk.

Finally, to enforce strict distributional alignment and ensure the Inter-Head produces accurate logits for direct decoding, we apply a stronger constraint on the interval's center prediction, $p_{\text{mid}}$. Its loss, $\mathcal{L}_c$, includes the anchoring loss plus an explicit KL-divergence term to match the cloud model's distribution: Distributional alignment is enforced through KL divergence regularization:

$$\mathcal{L}_{\text{align}} = \lambda_\beta D_{\text{KL}}(p_{\text{cloud}} \| p_{\text{mid}}) \tag{11}$$

The complete Inter-DRO loss integrates these components:

$$\begin{aligned}
\mathcal{L}_{\text{Inter-DRO}} =& \mathcal{L}_c + \mathcal{L}_u + \mathcal{L}_l \\
=& \underbrace{\mathcal{L}_{\text{anchor}}(\mathbf{p}_{\text{mid}}) + \mathcal{L}_{\text{align}}}_{\text{Center loss}} + \underbrace{\mathcal{L}_{\text{anchor}}(\mathbf{p}_{\text{up}})}_{\text{Upper bound}} + \underbrace{\mathcal{L}_{\text{anchor}}(\mathbf{p}_{\text{lo}}) + \mathcal{L}_{\text{lo}}^{\text{DRO}}}_{\text{Lower bound DRO}}
\end{aligned} \tag{12}$$

This entire training process is fully compatible with techniques such as Classifier-Free Guidance (CFG), which we employ to enhance sample diversity and quality.

Table 1: Main results. Our method significantly reduces latency and cloud requests while maintaining comparable CLIP score, FID, F1, and HPSv2 performance.

| Models | Methods | Metric | | | | | | |
|---|---|---|---|---|---|---|---|---|
| | | CLIP (↑) | FID (↓) | F1(↑) | HPSv2(↑) | Latency(s) | steps | Cloud Call |
| LlamaGen(Stage I) | Base | 0.3161 | 23.6900 | 0.6097 | 22.74 | x1.00 | x1.00 | 100.00% |
| | Eagle2 | 0.3115 | 25.0700 | 0.6017 | 22.74 | x1.02 | x1.16 | 87.02% |
| | Lantern | 0.3159 | 25.5494 | 0.5834 | 21.29 | x1.66 | x2.01 | 50.11% |
| | Entropy-Lens | 0.3132 | 24.5828 | 0.5796 | 22.03 | x1.70 | x2.05 | 52.34% |
| | CoDe (N = 0.3) | 0.2827 | 35.6670 | 0.4625 | 18.08 | x2.04 | x2.734 | 30.00% |
| | **Ours** | 0.3159 | 24.2459 | 0.5997 | 22.48 | **x2.53** | **x3.00** | **30.44%** |
| LlamaGen(Stage II) | Base | 0.2822 | 40.0709 | 0.5350 | 23.84 | x1.00 | x1.00 | 100.00% |
| | Eagle2 | 0.2925 | 40.9472 | 0.5334 | 23.18 | x1.03 | x1.19 | 84.55% |
| | Lantern | 0.2900 | 41.8766 | 0.5364 | 23.06 | x1.64 | x1.99 | 50.54% |
| | Entropy-Lens | 0.2869 | 46.7238 | 0.4713 | 22.10 | x1.91 | x2.44 | 41.29% |
| | CoDe (N = 0.3) | 0.2712 | 45.6822 | 0.4819 | 20.68 | x1.86 | x2.45 | 30.00% |
| | **Ours** | 0.2927 | 39.3085 | 0.5458 | 23.26 | **x2.13** | **x2.83** | **34.46%** |
| Anole | Base | 0.3215 | 19.9455 | 0.6544 | 23.52 | x1.00 | x1.00 | 100.00% |
| | Eagle2 | 0.3159 | 23.7103 | 0.6117 | 22.88 | x1.02 | x1.09 | 91.98% |
| | Lantern | 0.3181 | 23.9510 | 0.5969 | 22.92 | x1.25 | x1.81 | 50.35% |
| | Entropy-Lens | 0.2966 | 32.3533 | 0.5600 | 22.34 | x1.57 | x2.53 | 39.86% |
| | CoDe (N = 0.3) | 0.2781 | 36.7520 | 0.5597 | 21.94 | x1.55 | x2.89 | 30.00% |
| | **Ours** | 0.3171 | 23.8593 | 0.5970 | 23.14 | **x1.87** | **x3.29** | **29.88%** |

# 4 EXPERIMENTS

In this section, we empirically demonstrate the efficacy of CIAR. We first report the experimental setup. Then we evaluate CIAR with other baselines in perspective of image quality and acceleration. Finally, we conduct an ablation study to clarify the effectiveness of each component in our method.

## 4.1 EXPERIMENTAL SETUP

To validate CIAR, we conduct experiments on the text-conditioned LlamaGen-XL Stage I, LlamaGen-XL Stage II, and Anole as the cloud model. On the device side, we adopt the same architecture but restrict it to a single autoregressive layer for efficiency. We use MS-COCO (Lin et al., 2014) validation captions to generate images and compare with ground truth for evaluation.

For baselines, we consider: (i) **EAGLE-2** (Li et al., 2024), which achieves state-of-the-art performance in speculative decoding; (ii) **Lantern** (Jang et al., 2024), a strong visual speculative decoding method; (iii) **Entropy-lens** (Ali et al., 2025), which measures token-level uncertainty via entropy of transformer activations; (iv) **SoftmaxCorr** (Tu et al., 2024), which evaluates confidence using maximum softmax probability; (v) **Random-based** strategy following the baseline setup in STAR (Zhang et al., 2024); (vi) **CoDe** (Chen et al., 2025), an efficient collaborative acceleration method for VAR. For acceleration, we report speedup, device token acceptance rate, and cloud request rate. For image quality under acceleration, we adopt FID, CLIP score, F1 (Heusel et al., 2017; Hessel et al., 2021), and HPSv2 (Wu et al., 2023).Additionally, we conduct a quantitative analysis of image generation quality under acceleration. All model training, inference latency tests, and speedup evaluations were conducted on NVIDIA GeForce RTX 4090 GPU (24GB VRAM).

## 4.2 MAIN RESULTS

Table 1 compares CIAR against various baselines. We summarize the key observations as follows: (1) **Text SD does not work for visual task.** Applying speculative decoding designed for text yields poor results, providing almost no acceleration while noticeably degrading image quality. This is because image tokens exhibit strong distributional consistency, unlike text tokens, which contain richer information diversity; (2) **Existing visual acceleration methods remain suboptimal.** Lantern mandates universal cloud validation and consequently incurs high computational costs. Similarly, CoDe suffers from severe distribution shift when adapted to the Next-Token Prediction paradigm because the exclusive reliance on the small model for subsequent sequences compromises generation quality. (3) **Entropy-based method is inadequate for discrete visual tokens.** Augmenting Lantern

Table 2: Performance comparison of different uncertainty method.

| Models | Methods | Metric | | | | | | |
|---|---|---|---|---|---|---|---|---|
| | | CLIP (↑) | FID (↓) | F1(↑) | HPSv2(↑) | Speedup | Steps | Cloud Call |
| LlamaGen(Stage I) | Base | 0.3161 | 23.6900 | 0.6097 | 22.74 | x1.00 | x1.00 | 100.00% |
| | Random | 0.3142 | 30.1872 | 0.5369 | 18.16 | x2.28 | x2.36 | 36.46% |
| | Entropy-Lens | 0.3132 | 24.5828 | 0.5796 | 22.03 | x1.70 | x2.05 | 52.34% |
| | SoftmaxCorr | 0.3149 | 31.1009 | 0.5130 | 19.11 | x2.27 | x2.31 | 36.49% |
| | **Ours** | 0.3159 | 24.2459 | 0.5997 | 22.48 | **x2.53** | **x3.00** | **30.44%** |
| LlamaGen(Stage II) | Base | 0.2822 | 40.0709 | 0.5350 | 23.84 | x1.00 | x1.00 | 100.00% |
| | Random | 0.2876 | 44.9589 | 0.4629 | 20.76 | x2.09 | x2.48 | 34.58% |
| | Entropy-Lens | 0.2869 | 46.7238 | 0.4713 | 22.10 | x1.91 | x2.44 | 41.29% |
| | SoftmaxCorr | 0.2825 | 50.0306 | 0.4030 | 19.54 | x1.91 | x2.40 | 36.30% |
| | **Ours** | 0.2927 | 39.3085 | 0.5458 | 23.26 | **x2.13** | **x2.83** | **34.46%** |
| Anole | Base | 0.3215 | 19.9455 | 0.6544 | 23.52 | x1.00 | x1.00 | 100.00% |
| | Random | 0.2966 | 52.3533 | 0.3464 | 20.55 | x1.53 | x2.44 | 39.53% |
| | Entropy-Lens | 0.2966 | 32.3533 | 0.5600 | 22.34 | x1.57 | x2.53 | 39.86% |
| | SoftmaxCorr | 0.2966 | 51.0382 | 0.3447 | 20.96 | x1.48 | x2.30 | 41.04% |
| | **Ours** | 0.3171 | 23.8593 | 0.5970 | 23.14 | **x1.87** | **x3.29** | **29.88%** |

with standard entropy thresholds degrades performance because scalar entropy is too coarse to capture spatially heterogeneous and semantically critical ambiguities. (4) **CIAR achieves a superior efficiency–quality tradeoff.** Our method reduces inference latency by $3\times$ versus the base model with only 30% cloud request frequency and cuts the computational load of Lantern by 60% while improving CLIP and FID scores. These gains arise from the Inter-Head which produces calibrated probability intervals to drive a selective verification policy. This mechanism allows low-ambiguity regions to be generated locally while precisely offloading high-uncertainty tokens for cloud verification to preserve visual fidelity.

In addition, we conducted a visual analysis, as shown in Figure 4. The cases demonstrate that CIAR preserves fine details and maintains overall object coherence while reducing artifacts and distortions. The right side of the figure compares results with and without the Interval-Enhanced module, showing that collaboration further improves detail consistency and object integrity.

## 4.3 ANALYSIS OF DEVICE-SIDE UNCERTAINTY QUANTIFICATION

**Effect of Uncertainty Mechanism.** To evaluate the proposed Inter-Head mechanism, we compared it against Random, Entropy-lens, and SoftmaxCorr baselines as detailed in Table 2. The Random strategy significantly degrades FID while Entropy-lens fails to capture subtle visual differences among tokens. Similarly, SoftmaxCorr results in a sharp quality drop because relying on a single token probability ignores the broader distributional uncertainty. In contrast, our Inter-Head assesses uncertainty from the entire distribution and achieves a superior balance between cloud verification frequency and image fidelity.

**Effect of Device Model Capacity.** We further investigate the impact of device capacity including *Inter-Head capacity* and *device model depth* with detailed results provided in A.2.5 and A.2.4. Experiments on Inter-Head capacity using voting (Daliparthi, 2024) and ensemble (Duan et al., 2025) strategies show that increasing the head count to two or three improves image quality with negligible computational overhead, while adding further heads yields diminishing returns. Regarding model depth, we evaluated device models with varying layer counts. The results indicate that deeper device models enhance image quality and significantly reduce cloud interaction frequency, although this improvement comes at the cost of a slight increase in total inference latency.

## 4.4 ANALYSIS OF CLOUD-ENHANCED DECODING

**Effect of Prefix Rate.** We inject cloud-generated prefixes as distributional anchors to mitigate local generation drift. Figure 3 indicates that higher prefix rates steadily improve image quality and reduce the frequency of cloud verification requests. However, the inference speedup exhibits non-monotonic behavior because the computational cost of generating initial cloud tokens offsets the benefits of reduced verification at very high rates. A rate of 0.06 strikes an optimal balance between

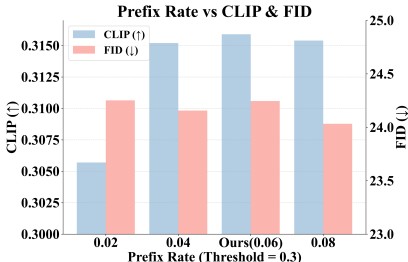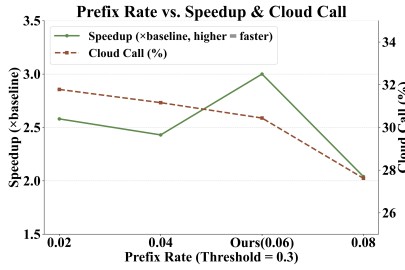

Figure 3: Comparison of different prefix rate

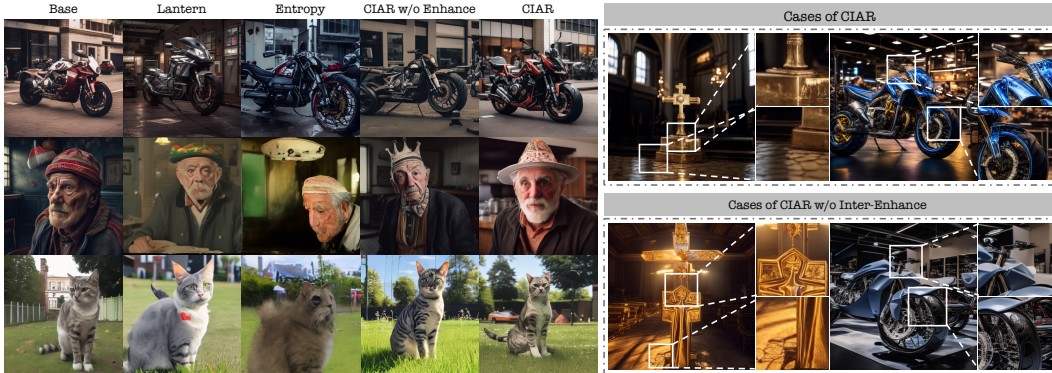

Figure 4: Visual analysis of different methods. "CIAR w/o Inter-Enhance" denotes our method without interval-enhanced decoding using interval features.

guidance and overhead whereas higher rates compromise latency. Visual comparisons in Figure 4 further confirm that our method yields superior consistency and realistic details.

**Effect of Uncertainty Threshold.** We further analyze the uncertainty threshold with full results in A.2.1. Increasing the threshold enhances speed by accepting more local tokens but risks quality degradation. We identify 0.30 as the optimal operating point which provides a significant speedup over conservative settings with negligible perceptual loss.

### 4.5 ANALYSIS OF CONTINUOUS-BASED UNCERTAINTY

Our continuous-based uncertainty estimation circumvents the exponential latency scaling inherent in discrete methods that require solution enumeration. As illustrated in Table 3 and Figure 5, discrete approaches incur prohibitive computational costs as the codebook size $k$ increases while yielding only marginal quality gains. In contrast, our method achieves significantly lower latency than the discrete counterpart at $k = 100$ while delivering superior CLIP and FID scores. These results confirm that continuous intervals provide a highly efficient estimation mechanism that ensures high-quality generation without the overhead of enumerating feasible solutions. Additional qualitative examples are detailed in A.2.2.

## 5 CONCLUSION

We introduced CIAR, a collaborative cloud-device framework for efficient autoregressive visual generation. At its core, CIAR introduces an interval-based uncertainty quantifier, Inter-Head, to enable on-device self-verification, reducing cloud computation. To maintain distributional consistency between cloud and device, CIAR leverages Cloud-Enhanced Decoding with interval-feature conditioning and prefix guidance, alongside an Inter-DRO loss for output alignment. Experiments demonstrate that CIAR achieves substantial speed-ups while preserving visual quality, enabling practical AR visual model deployment on devices and enhancing real-world multimodal user experiences.

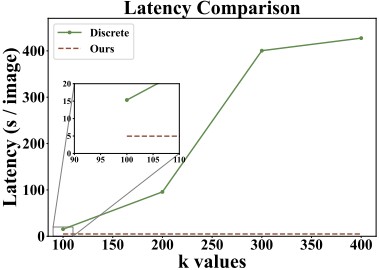

Figure 5: Latency comparison between Discrete and Continuous methods.

Table 3: Performance comparison between Discrete method and Continuous method.

| Methods | Settings | Metric | |
|---|---|---|---|
| | | CLIP (↑) | FID (↓) |
| Discrete | k = 100 | 0.3081 | 26.0371 |
| | k = 200 | 0.3096 | 25.5508 |
| | k = 300 | 0.3123 | 24.8237 |
| | k = 400 | 0.3091 | 24.9401 |
| Continuous | Ours | **0.3176** | **24.2463** |

## ACKNOWLEDGMENTS

National Natural Science Foundation of China (No. U24A20326, 62402429, 62441236). Supported by the Key Research and Development Program of Zhejiang Province (No. 2025C01026). Ningbo Yongjiang Talent Introduction Programme (2023A-397-G). Young Elite Scientists Sponsorship Program by CAST (2024QNRC001). The author gratefully acknowledges the support of Zhejiang University Education Foundation Qizhen Scholar Foundation. Sponsored by CIPS-LMG Huawei Innovation Fund.

## ETHICS STATEMENT

Our work on CIAR, a collaborative cloud-device framework for efficient autoregressive image generation, adheres to the ICLR Code of Ethics. The primary goal of CIAR is to enhance the reliability and transparency of visual synthesis systems through interval-based uncertainty quantification, which can benefit high-stakes applications such as medical imaging and assistive tools by reducing overconfidence and errors. In this study, no human subjects or animal experimentation was involved. All datasets used were sourced in compliance with relevant usage guidelines, ensuring no violation of privacy or security protocols. We have taken proactive measures to avoid biases in model training and evaluation, and no personally identifiable information was processed. While generative technologies like CIAR could potentially be misused for malicious purposes, such as creating deepfakes, we conducted this research with a commitment to responsible innovation, advocating for the development of safeguards like detection mechanisms. We maintain full transparency and integrity throughout the research process, aligning with ethical AI principles.

## REPRODUCIBILITY STATEMENT

To ensure the reproducibility of our work, we will open-source the implementation of the CIAR method. The complete code, along with detailed documentation, is included in the supplementary materials submitted with this paper. Additionally, upon acceptance, the code will be made publicly available on GitHub to facilitate further research and application.

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

# A  APPENDIX

## A.1  PSEUDO CODE

---

**Algorithm 1** CIAR: Collaborative Interval-based AutoRegressive Decoding

---

**Input:**

$\mathcal{M}_{\text{cloud}}$: Cloud autoregressive model
$\mathcal{M}_{\text{device}}$: Device autoregressive model
$\mathcal{H}_{\text{inter}}$: Interval-Head module
$\rho$: Prefix rate
$\tau$: Uncertainty threshold
$L$: Total sequence length
$K$: Device sequence length
prompt: Input conditioning

**Output:**  Generated token sequence $\mathbf{x}_{0:L-1}$

1: **Initialize:** $t \leftarrow 0$, $\mathbf{x} \leftarrow \emptyset$, **prefix** $\leftarrow \emptyset$
2: $m \leftarrow \lfloor \rho \cdot L \rfloor$                                                       ▷ Compute prefix length
3: **while** $t < m$ **do**
4:      $x_t \leftarrow \mathcal{M}_{\text{cloud}}(\text{prompt}, \mathbf{x}_{0:t-1})$              ▷ Auto-regressive generation
5:      $\mathbf{x} \leftarrow \mathbf{x} \cup \{x_t\}$, **prefix** $\leftarrow$ **prefix** $\cup \{x_t\}$
6:      $t \leftarrow t + 1$
7: **end while**
8: **while** $t < L$ **do**                        ▷ Device-side generation with verification
9:      $h_t \leftarrow \mathcal{M}_{\text{device}}(\textbf{prefix}, \mathbf{x}_{m:t-1})$             ▷ Device prediction
10:     $\mathbf{c}_t, \mathbf{r}_t \leftarrow \textbf{Inter-Head}(h_t)$             ▷ Interval center/radius
11:     $\mathcal{P}_t \leftarrow \textbf{InterFuse}(c_t, r_t)$
12:     $\mathcal{U}_t \leftarrow \mathcal{U}(\mathcal{P}_t)$                                 ▷ Eq. 4
13:     **if** $\mathcal{U}_t \leq \tau$ **then**
14:         Accept $x_t$                        ▷ On-device Self-verification
15:         $t \leftarrow t + 1$
16:     **else**
17:         **buffer** $\leftarrow \{x_t\}$, $k \leftarrow 1$
18:         **while** $k < K \wedge t + k < L$ **do**
19:             $f_{t+k}, \mathcal{P}_{t+k} \leftarrow \mathcal{M}_{\text{device}}(x_{0:t+k-1})$
20:             **buffer** $\leftarrow$ **buffer** $\cup (f_{t+k}, \mathcal{P}_{t+k})$
21:             $k \leftarrow k + 1$
22:         **end while**
23:         $f^I \leftarrow \phi(\textbf{Concat}(\textbf{buffer}))$                 ▷ Eq. 5
24:         $\mathbf{x}_{\text{verified}} \leftarrow \mathcal{M}_{\text{cloud}}(\mathbf{x}_{0:t+K})$          ▷ Cloud verification
25:         $\mathbf{x} \leftarrow \mathbf{x} \cup \mathbf{x}_{\text{verified}}$
26:         $x_{sample} \leftarrow \mathcal{M}_{cloud}(\mathbf{x}, f^I)$
27:         $t \leftarrow t + |\mathbf{x}_{\text{verified}}| + 1$
28:     **end if**
29: **end while**
            **return** $\mathbf{x}_{0:L-1}$

---

Algorithm 1 illustrates the complete inference process of the CIAR framework. Initially, the cloud model $M_{cloud}$ accepts an input prompt and generates a prefix sequence of image tokens. This prefix is then transferred to the device model $M_{device}$, which performs autoregressive generation. Each token produced by the device model is processed by the Inter-Head module to compute its uncertainty score. If the score surpasses a threshold $\tau$, the system collects a subsequent sequence of $K$ tokens and forwards them to the cloud for verification and resampling. During this step, the interval features derived from the tokens are also transmitted to the cloud as supplementary information. The autoregressive generation terminates once the total token sequence length reaches $L$, at which point image decoding begins.

## A.2 In-depth Analysis

### A.2.1 Analysis of different thresholds

We investigate the impact of the uncertainty threshold on generation quality and efficiency with a fixed prefix rate of 0.06. Figure 6 presents the corresponding CLIP, FID, speedup, and cloud call rate. Increasing the threshold allows more tokens to be accepted locally, which reduces cloud requests and improves inference speed. At a threshold of 0.30, the method achieves a CLIP score of 0.3159 and an FID of 24.2459, representing negligible perceptual differences compared to the more conservative threshold of 0.25, which yields 0.3154 and 24.2130. However, this slight adjustment brings a meaningful 15% relative speedup from 2.20× to 2.53× and lowers the cloud call rate from 30.98% to 30.44%. Further increasing the threshold beyond 0.30 leads to greater acceleration but significant degradation in visual quality. At 0.35, CLIP drops to 0.3108 and FID rises to 27.1990; at 0.40, CLIP falls to 0.3048 and FID worsens to 28.5853. These results show that blindly raising the threshold sacrifices image quality for speed. We select 0.30 as it offers a favorable balance between acceleration and perceptual quality.

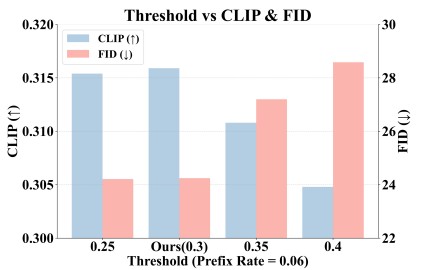 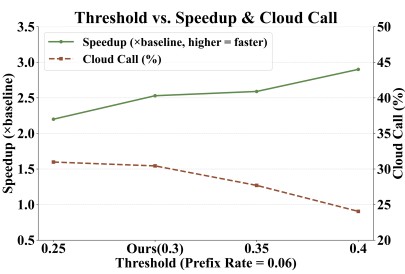

Figure 6: Comparison of different threshold

### A.2.2 Discrete vs. Continuous

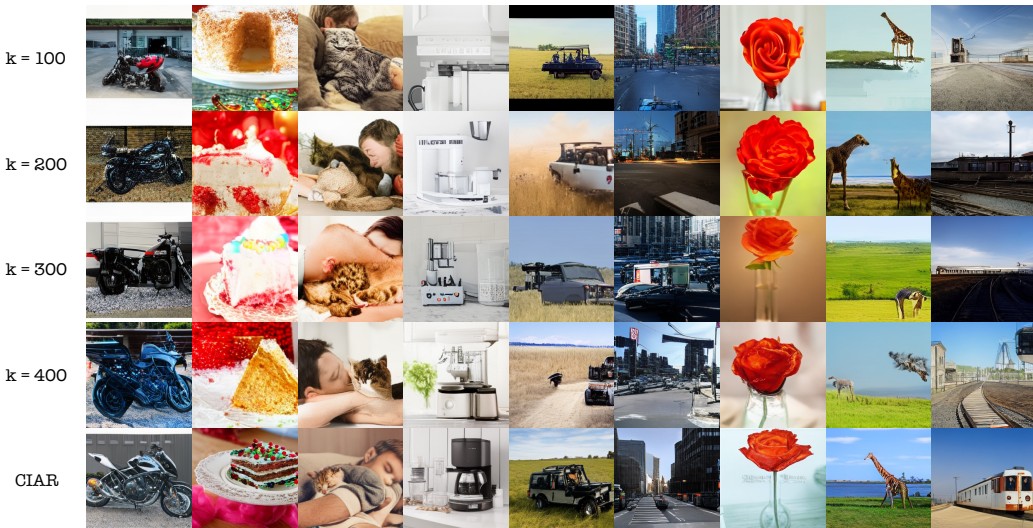

Figure 7: Visual analysis of Discrete metric and Continuous metric

Figure 7 presents image samples generated by the discrete uncertainty quantification method under varying k values. We evaluated four monotonically increasing k values and conducted a comparative analysis with the CIAR method using identical prompts and random seeds. The results indicate that at k=100, while latency is reduced, the limited token count for uncertainty computation leads to imprecise uncertainty estimates, resulting in the lowest image quality. As k increases, image details become more refined, yet significant distortions and artifacts remain apparent. In comparison,

Table 4: Impact of different $N$ values on CoDe and comparison with CIAR.

| Models | Method | Param. $N$ | Metric | | | | | |
|---|---|---|---|---|---|---|---|---|
| | | | CLIP ($\uparrow$) | FID ($\downarrow$) | F1($\uparrow$) | HPSv2($\uparrow$) | Latency(s) | Cloud Call |
| LlamaGen(Stage I) | Base | - | 0.3161 | 23.6900 | 0.6097 | 22.74 | x1.00 | 100.00% |
| | CoDe | 0.10 | 0.2566 | 44.7305 | 0.4028 | 15.84 | **x2.71** | 10.00% |
| | | 0.15 | 0.2620 | 42.6082 | 0.4088 | 16.36 | x2.41 | 15.00% |
| | | 0.20 | 0.2627 | 40.3943 | 0.4508 | 16.83 | x2.14 | 20.00% |
| | | 0.25 | 0.2783 | 38.3103 | 0.4639 | 17.43 | x2.13 | 25.00% |
| | | 0.30 | 0.2827 | 35.6670 | 0.4625 | 18.08 | x2.04 | 30.00% |
| | **Ours** | - | 0.3159 | 24.2459 | 0.5997 | 22.48 | x2.53 | 30.44% |

Table 5: Performance comparison of device models with varying depths.

| Models | #Layers | Metric | | | | | | |
|---|---|---|---|---|---|---|---|---|
| | | CLIP ($\uparrow$) | FID ($\downarrow$) | F1($\uparrow$) | HPSv2($\uparrow$) | Latency(s) | Steps | Cloud Call |
| LlamaGen(Stage I) | 1 Layer | 0.3159 | 24.2459 | 0.5997 | **22.48** | x2.53 | x3.00 | 30.44% |
| | 2 Layers | 0.3113 | 23.8820 | **0.6270** | 22.44 | **x2.54** | x3.44 | 27.06% |
| | 3 Layers | **0.3171** | 23.8404 | 0.5899 | 22.44 | x2.19 | x3.58 | 26.10% |
| | 5 Layers | **0.3171** | **23.8232** | 0.6180 | 22.45 | x2.14 | **x3.66** | **25.68%** |

the CIAR method achieves markedly superior image quality, with more accurate rendering of object details. This outcome underscores the ability of CIAR to effectively minimize latency while maintaining precise uncertainty quantification, thereby ensuring high image fidelity.

### A.2.3 COMPARISON WITH CODE IN NTP PARADIGM

CoDe (Chen et al., 2025) is a collaborative acceleration framework originally designed for the Next-Scale-Prediction (NSP) paradigm, assigning semantic-heavy early scales to the large model and subsequent scales to a smaller model. To adapt CoDe for LlamaGen (which follows the Next-Token-Prediction, NTP paradigm), we define $N$ as the ratio of the sequence predicted by the large cloud model. Table 4 presents the impact of $N$ on performance.

**Analysis.** While CoDe provides acceleration, it consistently underperforms CIAR in generation quality. This degradation stems from the fundamental difference between the paradigms: NSP predicts holistic image features where early scales provide a robust global context, whereas NTP relies on strict token-wise autoregression. When the device model (small model) predicts long consecutive sequences (low $N$), the lack of high-capacity correction leads to severe *distribution shift* and error accumulation. Increasing $N$ mitigates quality loss by involving the cloud model more frequently but significantly compromises the acceleration gain. Consequently, CoDe is suboptimal for NTP-based autoregressive models compared to CIAR's dynamic, uncertainty-aware allocation.

### A.2.4 IMPACT OF DEVICE MODEL LAYERS

We investigate the trade-off between the device model's capacity and overall efficiency by varying the device model depth (2, 3, and 5 layers). Results are summarized in Table 5.

**Analysis.** Increasing the network depth significantly enhances the device model's representational capability, resulting in improved FID scores. Crucially, a stronger device model generates tokens with higher confidence, thereby reducing the frequency of cloud model intervention (lower cloud call rate). However, this comes at the cost of increased computational latency per step on the device. The results suggest that slightly increasing the device model depth can be a viable strategy to balance generation quality and acceleration, depending on the specific hardware constraints.

### A.2.5 IMPACT OF INTER-HEAD CAPACITY

The Inter-Head is pivotal for uncertainty estimation. We examine the impact of its capacity by training 5 distinct Inter-Heads under different random seeds and aggregating them using two strategies:

Table 6: Impact of Inter-Head capacity with Ensemble and Voting aggregation mechanisms.

| Method | #Head | Metric | | | | | | |
|---|---|---|---|---|---|---|---|---|
| | | CLIP ($\uparrow$) | FID ($\downarrow$) | F1($\uparrow$) | HPSv2($\uparrow$) | Latency(s) | Steps | Cloud Call |
| Ensemble | 1 Head | 0.3159 | 24.2459 | 0.5997 | 22.48 | x2.53 | x3.00 | 30.44% |
| | 2 Heads | 0.3157 | 24.1517 | 0.5878 | 22.38 | x2.46 | x2.98 | 29.45% |
| | 3 Heads | 0.3151 | 24.1301 | 0.6099 | 22.30 | x2.45 | x2.97 | 28.86% |
| | 4 Heads | 0.3137 | 24.6195 | 0.5999 | 22.26 | x2.49 | x2.97 | 28.73% |
| | 5 Heads | 0.3147 | 24.6723 | 0.5988 | 22.26 | x2.50 | x3.00 | 28.38% |
| Voting | 1 Head | 0.3159 | 24.2459 | 0.5997 | 22.48 | x2.53 | x3.00 | 30.44% |
| | 2 Heads | 0.3167 | 24.0400 | 0.6095 | 22.34 | x2.29 | x2.98 | 29.38% |
| | 3 Heads | 0.3165 | 24.1466 | 0.6195 | 22.25 | x2.48 | x3.00 | 28.97% |
| | 4 Heads | 0.3169 | 24.3542 | 0.5678 | 22.25 | x2.53 | x2.98 | 28.56% |
| | 5 Heads | 0.3135 | 24.3353 | 0.5910 | 22.22 | x2.48 | x2.98 | 28.54% |

Table 7: Ablation study on the Inter-DRO loss.

| Models | Loss | Metric | | | | | | |
|---|---|---|---|---|---|---|---|---|
| | | CLIP ($\uparrow$) | FID ($\downarrow$) | F1($\uparrow$) | HPSv2($\uparrow$) | Latency(s) | Steps | Cloud Call |
| LlamaGen(Stage I) | w/o $L_c$ | 0.3149 | 25.2721 | 0.5889 | 21.92 | x2.37 | x2.98 | 28.05% |
| | w/o $L_u$ | 0.2891 | 47.8981 | 0.4110 | 16.65 | x2.39 | x2.99 | 32.17% |
| | w/o $L_1$ | 0.2945 | 25.2849 | 0.5889 | 22.45 | x2.37 | x2.98 | 28.05% |
| | w/o $L_{\text{anchor}}$ | 0.2168 | 68.6400 | 0.4114 | 15.78 | x2.30 | x2.85 | 33.79% |
| | **Ours** | 0.3159 | 24.2459 | 0.5997 | 22.48 | **x2.53** | **x3.00** | 30.44% |

- **Voting (Daliparthi, 2024):** A hard-voting mechanism where the final output is determined by the majority class prediction of the heads, prioritizing robustness.

- **Ensemble (Duan et al., 2025):** A soft-voting mechanism that averages the probability distributions from multiple heads to obtain a calibrated score.

**Analysis.** As shown in Table 6, increasing the number of heads has a negligible impact on latency due to the lightweight nature of the Inter-Head. Performance in terms of FID peaks when using 2 or 3 heads. However, increasing the count to 5 leads to a degradation in CLIP, FID, and HPSv2 scores across both strategies. This indicates that excessive capacity in the uncertainty estimation module does not yield better calibration and may lead to overfitting or noise accumulation.

### A.2.6 IMPACT OF THE INTER-DRO LOSS

We conduct a comprehensive ablation study to isolate the contributions of each Inter-DRO component, specifically excluding the anchor loss to verify its role in device-cloud alignment. As detailed in Table 7, the anchor loss $\mathcal{L}_{\text{anchor}}$ proves fundamental for guaranteeing the classification accuracy of the Inter-Head, and its removal leads to poor training convergence alongside a marked degradation in image quality. Regarding the core Inter-DRO terms, the center loss $\mathcal{L}_c$ aligns the device-cloud distributions via KL divergence, aiming for soft probability matching rather than the hard accuracy enforcement of the anchor loss. Simultaneously, the lower-bound loss $\mathcal{L}_l$ facilitates the detection of uncertain tokens. Ablating either $\mathcal{L}_c$ or $\mathcal{L}_l$ results in a minor decline in collaborative performance. In contrast, the upper-bound loss $\mathcal{L}_u$ is critical for the precise selection of target tokens, evidenced by the significant quality drop upon its removal. Collectively, these components empower the Inter-Head to accurately discern valid tokens and robustly evaluate uncertainty.

### A.2.7 SIMULATION OF COMMUNICATION COSTS

Due to physical resource constraints, we simulate the cloud-device collaboration environment following standard protocols in cloud-device collaboration works (Kang et al., 2017; Laskaridis et al., 2020; Li et al., 2018; Lv et al., 2023a). We evaluate the inference performance across 5G, 4G, and WiFi network conditions.

Table 8: Comparison of communication costs between Lantern and CIAR under different network conditions. **Device** and **Cloud** denote the computational latency on the respective endpoints, while **Communication** represents the data transmission overhead.

| Network | Method | Device (ms) | Cloud (ms) | Communication (ms) | Total (ms) | Ratio (Comm/Tot) |
|---|---|---|---|---|---|---|
| 5G | Lantern | 11.63 | 11586.29 | 4424.63 | 16022.54 | 0.2762 |
|  | **Ours** | 13.54 | 5102.6 | **1678.13** | **6794.27** | **0.247** |
| 4G | Lantern | 11.9 | 11577.91 | 22291.31 | 33881.12 | 0.6579 |
|  | **Ours** | 14.1 | 4847.2 | **8173.95** | **13035.26** | **0.6271** |
| WiFi | Lantern | 11.69 | 11436.11 | 8768.01 | 20215.81 | 0.4337 |
|  | **Ours** | 14.11 | 5052.52 | **3343.94** | **8410.58** | **0.3976** |

**Simulation Metric.** The total latency $T_{\text{total}}$ incorporates both communication and computational overhead. The communication latency $T_{\text{comm}}$ for a single transmission event is calculated as:

$$T_{\text{comm}} = L + \frac{DataSize}{B} \tag{13}$$

where $L$ denotes the network latency (RTT), $B$ represents the bandwidth, and $DataSize$ refers to the transmission payload size. Consequently, the total inference latency is formulated as the sum of bidirectional communication and computation costs:

$$T_{\text{total}} = T_{\text{comm}}^{\text{cloud}} + T_{\text{comm}}^{\text{device}} + T_{\text{cloud}} + T_{\text{device}} \tag{14}$$

Here, $T_{\text{cloud}}$ and $T_{\text{device}}$ represent the computational latency on the cloud and device, respectively, while $T_{\text{comm}}^{\text{cloud}}$ and $T_{\text{comm}}^{\text{device}}$ denote the downlink (cloud-to-device) and uplink (device-to-cloud) transmission latencies. Specific network configurations are detailed in Table 9.

**Analysis.** As shown in Table 8, CIAR demonstrates superior bandwidth efficiency compared to the baseline Lantern method. By accurately identifying trustworthy tokens locally, CIAR minimizes the transmission payload, resulting in a significantly lower Communication-to-Total Latency ratio (e.g., **0.247 vs. 0.276** under 5G). This selective transmission strategy effectively offloads computational burdens from the cloud while mitigating network bottlenecks.

Table 9: Detailed configurations of different network conditions.

| Network | Bandwidth (Mbps) | Latency (ms) |
|---|---|---|
| 5G | 300.00 | 10.00 |
| 4G | 20.00 | 50.00 |
| WiFi | 100.00 | 20.00 |

### A.2.8 GENERALIZATION TO DIFFERENT ARCHITECTURES

To evaluate the architectural universality of CIAR, we extend our experiments to two distinct generative frameworks on ImageNet-1K: VAR (based on Next-Scale Prediction) and DiT (a Diffusion-based model).

**Settings.** For the VAR adaptation, we utilize a single autoregressive layer to predict the next scale, equipped with a fine-tuned Inter-Head. For DiT, we adapt a single DiT block equipped with an Inter-Head to predict noise and variance for the subsequent timestep.

**Analysis.** In the VAR context, CIAR successfully accelerates inference while preserving image fidelity. However, due to the inherently shorter sequence lengths in the NSP paradigm, the relative overhead of cloud verification becomes more significant, resulting in a slightly reduced acceleration gain compared to CoDe in this specific setting. Conversely, for diffusion models like DiT, CIAR demonstrates remarkable efficiency, achieving IS and FID scores comparable to a 25-step DDIM sampler but with a **2.5× speedup**. This underscores the potential of CIAR in accelerating iterative refinement processes beyond standard autoregression.

### A.3 ANALYSIS OF THE UNCERTAINTY QUANTIFICATION

In this section we analyze, refine, and justify the uncertainty score given in Eq. equation 4. We denote by $n := |\mathcal{V}|$ the vocabulary size, and for brevity drop the time index $t$ when the derivation is

Table 10: Performance of CIAR applied to VAR and DiT architectures on ImageNet-1K.

| Model | Method | IS↑ | FID↓ | Precision↑ | Recall↑ | Latency |
|---|---|---|---|---|---|---|
| DiT | DDIM-50 steps | 241 | 2.27 | - | - | x1.00 |
| | DDIM-25 steps | 232 | 3.18 | - | - | x2.00 |
| | DDIM-20 steps | 221 | 3.81 | - | - | x2.50 |
| | DDIM-12 steps | 184 | 7.80 | - | - | x4.17 |
| | **Ours** | 240 | 2.32 | - | - | **x4.90** |
| VAR | VAR-d24 | 311 | 2.11 | 0.82 | 0.59 | x1.00 |
| | VAR-d20 | 301 | 2.61 | 0.83 | 0.56 | x1.65 |
| | CoDe | 297 | 2.26 | 0.80 | 0.60 | **x2.90** |
| | **Ours** | 290 | 2.34 | 0.81 | 0.60 | x2.60 |

pointwise for a single autoregressive step. Thus

$$\delta = (\delta_1, \ldots, \delta_n)^\top \in \mathbb{R}_{\geq 0}^n, \qquad \delta_i := p_i^u - p_i^\ell, \tag{15}$$

$$\Omega := \|\delta\|_1 = \sum_{i=1}^n \delta_i, \qquad \bar{\delta} := \frac{\Omega}{n}, \qquad \Sigma := \sqrt{\frac{1}{n} \sum_{i=1}^n (\delta_i - \bar{\delta})^2}, \tag{16}$$

and the uncertainty score is

$$\mathcal{U}(\mathcal{P}) = \Omega \cdot \Sigma. \tag{17}$$

### A.3.1 PRINCIPLES: STATEMENT, CHECKING AND MINOR REFINEMENT

First, based on the characteristics of visual autoregressive generation, we propose three qualitative principles, followed by a concise mathematical clarification.

**Principle 1 (Total Ambiguity).** Uncertainty increases with the total volume of the probability space.

$$\sum_i (q_i^U - q_i^L) \uparrow \quad \Rightarrow \quad \mathcal{U} \uparrow \tag{18}$$

*Clarification.* The total interval volume is measured by $\Omega = \sum_i \delta_i$. Intuitively $\Omega$ upper-bounds the $L_1$-diameter of the feasible probability polytope (see Proposition 1 below), hence it is a natural first-order measure of how much the predicted distribution may vary.

**Principle 2 (Confidence Disparity).** Uncertainty is also high when confidence varies greatly across tokens (some intervals wide, others narrow). Uniformly narrow intervals imply low uncertainty.

$$\mathrm{Var}\big(q_i^U - q_i^L\big) \uparrow \quad \Rightarrow \quad \mathcal{U} \uparrow \tag{19}$$

*Clarification.* The standard deviation $\Sigma$ quantifies the spread (anisotropy) of allowable perturbations across coordinates. A large $\Sigma$ indicates the ambiguity is concentrated in a subset of tokens (directional uncertainty), whereas $\Sigma = 0$ indicates perfectly uniform widths $\delta_i = \bar{\delta}$ for all $i$. We emphasize an important design decision: the proposed metric intentionally *combines* total volume and disparity so that the score is elevated only when both the total freedom *and* its anisotropy are significant. This is consistent with downstream needs in autoregressive visual synthesis, where one often wants to detect "actionable" uncertainty (large and structured ambiguity) rather than ubiquitous, perfectly symmetric fuzziness across the whole vocabulary.

**Principle 3 (Local Certainty).** If probability mass is concentrated on a single token (an interval vanishes and its upper bound approaches 1), uncertainty diminishes.

$$\exists i: \ (q_i^U - q_i^L) \to 0 \ \wedge \ q_i^U \to 1 \quad \Rightarrow \quad \mathcal{U} \downarrow \tag{20}$$

*Clarification.* If a token $i$ attains (approximately) all probability mass in both lower and upper bounds (i.e., $p_i^\ell \approx p_i^u \approx 1$), then necessarily the remaining tokens' feasible probability mass becomes (nearly) zero. Under this natural consistency regime the widths $\delta_j$ for $j \neq i$ must be small and thus $\Omega$ and $\Sigma$ are small, yielding small $\mathcal{U}$. We make precise statements of this type below (Proposition 2).

### A.3.2 FROM PRINCIPLES TO THE PRODUCT FORM

We now justify the particular algebraic choice $\mathcal{U} = \Omega \cdot \Sigma$.

**Axiomatic Desiderata.** A simple set of desiderata for a scalar function $f(\delta)$ capturing the following principles:

1. $f(\delta) \geq 0$ for all $\delta$;
2. $f(\delta) = 0$ if $\Omega = 0$ (no ambiguity) or if the widths are perfectly uniform ($\Sigma = 0$), reflecting the design decision that perfectly symmetric ambiguity is not considered "high priority" uncertainty for autoregressive decision-making;
3. $f(\alpha\delta)$ should increase with scalar $\alpha > 1$ (monotonicity in total scale);
4. $f$ should be sensitive to both scale ($\Omega$) and dispersion ($\Sigma$) and be simple to compute and interpret.

These requirements are satisfied by the multiplicative ansatz $f(\delta) = g(\Omega) \cdot h(\Sigma)$ with monotone $g, h$ and the simplest choice $g(\Omega) = \Omega$, $h(\Sigma) = \Sigma$. The product has the desirable property that it vanishes if either factor is zero and scales quadratically under uniform scaling $\delta \mapsto \alpha\delta$, which matches the intuition that both a large feasible set *and* substantial anisotropy are needed to produce a high-priority uncertainty signal.

### A.3.3 MATHEMATICAL PROPERTIES AND PROOFS

We now give several formal properties of $\mathcal{U}$.

**Claim 1** (Non-negativity and Zeros). *$\mathcal{U}(\mathcal{P}) \geq 0$. Moreover $\mathcal{U}(\mathcal{P}) = 0$ if and only if $\Omega = 0$ or $\Sigma = 0$ (i.e., $\delta$ is the all-zero vector or $\delta$ is constant across coordinates).*

*Proof.* Immediate from definitions: $\Omega \geq 0$, $\Sigma \geq 0$, hence $\mathcal{U} = \Omega\Sigma \geq 0$. If $\Omega = 0$ then $\delta \equiv 0$ and $\Sigma = 0$ and $\mathcal{U} = 0$. If $\Sigma = 0$ then all $\delta_i = \bar{\delta}$ and $\mathcal{U} = \Omega \cdot 0 = 0$. Conversely, $\Omega\Sigma = 0$ implies either $\Omega = 0$ or $\Sigma = 0$. $\quad\square$

**Claim 2** (Scaling). *For any scalar $\alpha \geq 0$, $\mathcal{U}(\alpha\delta) = \alpha^2 \mathcal{U}(\delta)$.*

*Proof.* If $\delta' = \alpha\delta$ then $\Omega' = \alpha\Omega$ and $\Sigma' = \alpha\Sigma$; multiply to obtain $\mathcal{U}' = \alpha^2 \mathcal{U}$. $\quad\square$

**Claim 3** (Upper Bound in Terms of $\Omega$). *For any $\delta \geq 0$,*

$$\Sigma \leq \frac{\sqrt{n-1}}{n}\Omega, \qquad \text{and hence} \qquad \mathcal{U} \leq \frac{\sqrt{n-1}}{n}\Omega^2. \tag{21}$$

*The upper bound is tight: equality is attained when one coordinate equals $\Omega$ and the remaining $n-1$ coordinates are zero.*

*Proof.* For fixed $\Omega$ the quantity $\sum_i \delta_i^2$ is maximized by concentrating all mass on a single coordinate (this is the well-known extremal property of $\ell_2$ under fixed $\ell_1$). Concretely, with $\bar{\delta} = \Omega/n$,

$$\Sigma^2 = \frac{1}{n}\sum_{i=1}^n \delta_i^2 - \bar{\delta}^2 \leq \frac{1}{n}\Omega^2 - \frac{\Omega^2}{n^2} = \frac{\Omega^2}{n^2}(n-1).$$

Taking square-roots yields the bound on $\Sigma$; multiplying by $\Omega$ gives the stated bound on $\mathcal{U}$. The one-hot vector $(\Omega, 0, \ldots, 0)$ attains the bound. $\quad\square$

**Claim 4** (Upper Bound in Terms of $\sum \delta_i^2$). *Let $S_2 := \sum_{i=1}^n \delta_i^2$. Then*

$$\mathcal{U} \leq \frac{S_2}{2}. \tag{22}$$

*This bound is tight for a suitable two-point configuration.*

*Proof.* Write $\mu = \bar{\delta}$. Then $S_2 = n(\Sigma^2 + \mu^2)$. Using $\mathcal{U} = n\mu\Sigma$ and treating $S_2$ as fixed, maximize $g(\mu) := n\mu\sqrt{S_2/n - \mu^2}$ over $\mu \in [0, \sqrt{S_2/n}]$. Standard calculus yields the maximizer $\mu^2 = S_2/(2n)$ and the maximum value $g_{\max} = S_2/2$. $\qquad\square$

**Proposition 1** (Feasible Set Diameter). *Let $\mathcal{P} := \{p \in \mathbb{R}^n : p_i^\ell \leq p_i \leq p_i^u, \ \sum_i p_i = 1\}$ be the feasible polytope of true discrete distributions consistent with the interval bounds. Then for any $p, q \in \mathcal{P}$,*

$$\|p - q\|_1 \leq \Omega. \tag{23}$$

*Consequently $\Omega$ upper-bounds the $L_1$-diameter of $\mathcal{P}$.*

*Proof.* For each coordinate $i$ we have $|p_i - q_i| \leq \delta_i$ because both $p_i, q_i$ lie in the interval $[p_i^\ell, p_i^u]$. Summing yields $\|p - q\|_1 = \sum_i |p_i - q_i| \leq \sum_i \delta_i = \Omega$. $\qquad\square$

**Proposition 2** (Local Certainty (A Sufficient Condition)). *Suppose there exists an index $k$ such that $p_k^\ell \to 1$ and $p_k^u \to 1$. Then $\Omega \to 0$ and hence $\mathcal{U} \to 0$. More generally, if $p_k^\ell \to 1$ and $\sum_{j \neq k} p_j^u \to 0$, then $\Omega \to 0$.*

*Proof.* If $p_k^\ell \to 1$ then, because $\sum_i p_i^\ell \leq 1$, necessarily $p_j^\ell \to 0$ for all $j \neq k$. If also $p_k^u \to 1$ and $\sum_{j \neq k} p_j^u \to 0$, then for $j \neq k$ both $p_j^u \to 0$ and $p_j^\ell \to 0$. Hence for all $j \neq k$, $\delta_j = p_j^u - p_j^\ell \to 0$, and $\delta_k = p_k^u - p_k^\ell \to 0$ by the hypothesis $p_k^u, p_k^\ell \to 1$. Therefore $\Omega = \sum_i \delta_i \to 0$, and since $\Sigma$ is bounded by $\Omega$ up to constants, $\mathcal{U} = \Omega\Sigma \to 0$. $\qquad\square$

### A.3.4 STATISTICAL / GEOMETRIC INTERPRETATION

The feasible set $\mathcal{P}$ is the intersection of the probability simplex with the axis-aligned hyperrectangle $\prod_i [p_i^\ell, p_i^u]$. Two geometric consequences follow.

- **Diameter control.** By Proposition 1, $\Omega$ upper-bounds the maximal $L_1$ distance between feasible distributions. Thus $\Omega$ controls the worst-case change in any linear functional $a^\top p$ over $\mathcal{P}$: for any $a \in \mathbb{R}^n$,

$$\max_{p,q \in \mathcal{P}} |a^\top p - a^\top q| \leq \|a\|_\infty \max_{p,q} \|p - q\|_1 \leq \|a\|_\infty \Omega. \tag{24}$$

  In particular, the range of any coordinate $p_i$ is bounded by $\delta_i$ and the range of expectations of bounded functions is controlled by $\Omega$.

- **Anisotropy and actionability.** $\Sigma$ quantifies the anisotropy of the hyperrectangle $\{\delta_i\}$ (it is zero for an isotropic rectangle $\delta_i = \text{const}$). High anisotropy (large $\Sigma$) implies that the feasible polytope allows comparatively large perturbations along a small set of coordinates—this is precisely the case where one can expect actionable model decisions (e.g. choose to re-query or defer) centred on particular tokens.

The chosen product $\mathcal{U} = \Omega\Sigma$ therefore measures (i) how large the feasible set can be (through $\Omega$) and (ii) how anisotropic/localized that freedom is (through $\Sigma$). Both ingredients are important for an uncertainty signal that feeds a selective acceptance/deferral policy in autoregressive synthesis.

### A.3.5 FURTHER ALGEBRAIC REWRITES AND INTUITION

Let $\mu = \bar{\delta} = \Omega/n$ and define the coefficient of variation

$$\text{CV} := \frac{\Sigma}{\mu} \tag{25}$$

whenever $\mu > 0$. Then

$$\mathcal{U} = n\mu\Sigma = n\mu^2 \, \text{CV}. \tag{26}$$

This decomposition clarifies trade-offs:

- If the average width $\mu$ is tiny, $\mathcal{U}$ is small even when relative dispersion CV is large.
- If CV is tiny (nearly uniform widths), $\mathcal{U}$ is small regardless of $\mu$.
- If both $\mu$ and CV are moderate-to-large, $\mathcal{U}$ grows quickly (quadratically in the uniform-scaling sense).

### A.3.6 PRACTICAL REMARKS FOR THRESHOLDING AND NORMALIZATION

1. *Threshold selection.* Because $\mathcal{U}$ scales quadratically under $\delta \mapsto \alpha\delta$, raw thresholds should account for the scale of typical interval widths. In practice one can either (i) normalize by $n$ or by $\Omega^2$ to obtain a dimensionless metric, or (ii) calibrate the threshold on a held-out validation set of generation situations.

2. *Behavior for perfectly uniform large widths.* As designed, $\mathcal{U}$ vanishes when widths are perfectly uniform. If an application requires a non-zero response for the case "all tokens are equally very ambiguous", replace the multiplicative factor $\Sigma$ by $\sqrt{\Sigma^2 + \varepsilon}$ or use a combined additive form $\lambda_1\Omega + \lambda_2\Omega\Sigma$ with small $\lambda_1 > 0$. This is a deliberate modeling choice; the product form emphasizes structured (non-isotropic) ambiguity.

3. *Computational cost.* $\Omega$ and $\Sigma$ are $O(n)$ to compute and therefore inexpensive even for large vocabularies.

### A.3.7 SUMMARY

The proposed metric $\mathcal{U}(\mathcal{P}) = \Omega\Sigma$ (i) is nonnegative and simple to compute, (ii) vanishes in the natural limiting regimes (no ambiguity, or perfectly uniform ambiguity), (iii) scales quadratically under uniform dilation of widths, (iv) admits tight bounds in terms of $\Omega$ and $\sum \delta_i^2$, and (v) admits a clear geometric/probabilistic interpretation via the feasible polytope $\mathcal{P}$. Together these properties rigorously support the design principles and show why $\mathcal{U}$ is an effective, low-cost scalar proxy for actionable uncertainty in autoregressive visual synthesis.

$\square$

### A.4 PROOF OF INTERVAL VALIDITY FOR INTER-HEAD OUTPUT

The Inter-Head module outputs a center vector $\mathbf{c}_t$ and a radius vector $\mathbf{r}_t$ for the hidden state $\mathbf{h}_t$, where $\mathbf{r}_t$ is non-negative due to the Softplus function. The logit interval for each token $i$ is defined as $[l_i^l, l_i^u] = [c_i - r_i, c_i + r_i]$. To convert these logit intervals into probability intervals $\mathcal{P}_t = [p_i^l, p_i^u]$, we employ the **InterFuse** operator, which ensures that the resulting probability intervals satisfy the validity condition $\sum_i p_i^l \leq 1 \leq \sum_i p_i^u$.

The **InterFuse** operator computes the lower and upper probability bounds for each token $i$ as follows:

$$p_i^l = \frac{\exp(c_i - r_i)}{\sum_j \exp(c_j) - \exp(c_i) + \exp(c_i - r_i)}, \tag{27}$$

$$p_i^u = \frac{\exp(c_i + r_i)}{\sum_j \exp(c_j) - \exp(c_i) + \exp(c_i + r_i)}. \tag{28}$$

These expressions ensure that $0 \leq p_i^l \leq p_i^u \leq 1$ for each token $i$, as the denominators are constructed to be greater than or equal to the numerators. However, the sums $S_l = \sum_i p_i^l$ and $S_u = \sum_i p_i^u$ may not strictly satisfy $S_l \leq 1 \leq S_u$ due to the nonlinear interactions between the terms.

To guarantee the interval condition, we introduce a normalization step. Let $S_l$ and $S_u$ be the sums after the initial computation. We scale the probabilities as:

$$p_i^l \leftarrow p_i^l \cdot \min\left(1, \frac{0.99}{S_l}\right), \tag{29}$$

$$p_i^u \leftarrow p_i^u \cdot \max\left(1, \frac{1.01}{S_u}\right). \tag{30}$$

This scaling ensures that $\sum_i p_i^l \leq 0.99 \leq 1$ and $\sum_i p_i^u \geq 1.01 \geq 1$, thus strictly satisfying the interval requirement. The constants 0.99 and 1.01 provide a margin for numerical stability and are chosen empirically.

In the code implementation, we further enhance numerical stability by using the max-trick (subtracting the maximum value of $\mathbf{c}_t$ before exponentiation) and clamping the radius to a maximum value. The final algorithm ensures that the probability intervals are valid and suitable for downstream tasks such as uncertainty quantification and collaborative decoding.

## A.5 USE OF LARGE LANGUAGE MODELS (LLMS)

During the initial exploratory phase of this research, we employed large language models solely to assist in retrieving relevant literature on autoregressive image generation. It is important to note that the LLM was not involved in the ideation, writing, or experimental design. All research concepts, ideas, and analyses were developed and conducted by the authors.

