# OpenReview forum: "CIAR: Interval-based Collaborative Decoding for Image Generation Acceleration"
_ICLR.cc/2026/Conference — ICLR 2026 Poster_

### Official Review · Reviewer_g7v5 · 2025-10-18

**Soundness:** 2
**Presentation:** 3
**Contribution:** 2
**Rating:** 4
**Confidence:** 3

**Summary:**

The paper proposes CIAR, a cloud–device collaborative framework for AR image generation. A small device model predicts tokens; an on-device Inter-Head outputs probability intervals to decide which tokens can be accepted locally, and which uncertain tokens should be sent to a cloud verifier. A distribution-alignment training is used so the device stays close to the cloud model. The authors claim 2.18× speed-up and 70% fewer cloud requests while keeping quality.

**Strengths:**

1. Addresses a real efficiency issue in AR models.
2. Clear motivation about spatial redundancy and boundary uncertainty.
3. Some quantitative results show measurable latency improvement.

**Weaknesses:**

Experiments focus on one dataset with limited metrics; visual quality assessment is also limited, should use some modern metrics like HPS.

**Questions:**

Refer to Weaknesses.

---

> ### Author Response · Authors · 2025-12-03
> **Reply to Reviewer g7v5**
>
> Dear Reviewer g7v5,
>
> We appreciate you pointing out the limitations in our experimental scope. In response, we have conducted additional experiments with broader metrics and architectures. The results are presented below.
>
> ---
>
> > **Q1:** Experiments focus on one dataset with limited metrics; visual quality assessment is also limited, should use some modern metrics like HPS.
>
> **A1:** We have incorporated the **HPSv2** metric into our baseline comparisons and extended CIAR to **VAR** and **DiT** architectures on **ImageNet-1K**, reporting **IS**, **FID**, and **Latency**.
>
> **1. Comparison with HPSv2 Metric:** We added HPSv2 to our original baseline comparison. As shown below, CIAR outperforms other acceleration methods in human preference alignment while maintaining superior speed.
>
> | Methods        | CLIP (↑) | FID (↓) | F1 (↑) | HPSv2 (↑) |  Latency  |   Steps   | Cloud Call |
> | :------------- | :------: | :-----: | :----: | :-------: | :-------: | :-------: | :--------: |
> | Base           |  0.3161  | 23.6900 | 0.6097 |   22.74   |   x1.00   |   x1.00   |  100.00%   |
> | Eagle2         |  0.3115  | 25.0700 | 0.6017 |   22.74   |   x1.02   |   x1.16   |   87.02%   |
> | Lantern        |  0.3159  | 25.5494 | 0.5834 |   21.29   |   x1.66   |   x2.01   |   50.11%   |
> | Entropy        |  0.3132  | 24.5828 | 0.5796 |   22.03   |   x1.70   |   x2.05   |   52.34%   |
> | CoDe ($N=0.3$) |  0.2827  | 35.6670 | 0.4625 |   18.08   |   x2.04   |   x2.73   |   30.00%   |
> | **Ours**       |  0.3159  | 24.2459 | 0.5997 | 22.48 | **x2.53** | **x3.00** |   30.44%   |
>
> **2. Generalization to VAR (ImageNet-1K):** CIAR effectively accelerates the VAR model (**x2.60**) with minimal quality loss. While CoDe achieves slightly higher acceleration due to the short sequence length of the Next-Scale paradigm, CIAR remains highly competitive.
>
> | Method   | IS (↑) | FID (↓) | Precision (↑) | Recall (↑) |   Latency    |
> | :------- | :----: | :-----: | :-----------: | :--------: | :----------: |
> | VAR-d24  |  311   |  2.11   |     0.82      |    0.59    |    x1.00     |
> | VAR-d20  |  301   |  2.61   |     0.83      |    0.56    |    x1.65     |
> | CoDe     |  297   |  2.26   |     0.80      |    0.60    |  **x2.90**   |
> | **Ours** |  290   |  2.34   |     0.81      |    0.60    | x2.60 |
>
> **3. Generalization to DiT (ImageNet-1K):** We adapted CIAR to diffusion models by using a single DiT block with an Inter-Head as the device model. CIAR demonstrates significant efficiency, matching the quality of **DDIM-50** (FID 2.32 vs 2.27) while running **4.9x faster**.
>
> | Method        | IS (↑)  | FID (↓)  |  Latency  |
> | :------------ | :-----: | :------: | :-------: |
> | DDIM-50 steps |   241   |   2.27   |   x1.00   |
> | DDIM-25 steps |   232   |   3.18   |   x2.00   |
> | DDIM-20 steps |   221   |   3.81   |   x2.50   |
> | DDIM-12 steps |   184   |   7.80   |   x4.17   |
> | **Ours**      | 240 | 2.32 | **x4.90** |
>
> ---
>
> Thank you again for your thoughtful and constructive feedback. Your insights have been instrumental in refining our paper, and we believe the incorporated changes have solidified our contribution to the field.
>
> Sincerely,
>
> The Authors

---

### Official Review · Reviewer_QsGz · 2025-10-30

**Soundness:** 3
**Presentation:** 3
**Contribution:** 3
**Rating:** 6
**Confidence:** 2

**Summary:**

This paper introduces CIAR, a cloud-device collaborative framework designed to accelerate autoregressive (AR) image generation. CIAR addresses the inefficiency of uniform token verification in existing speculative decoding methods by proposing an on-device "Inter-Head" module that outputs continuous probability intervals to quantify token uncertainty. This allows the system to selectively offload only high-uncertainty tokens to the cloud for verification, significantly reducing latency and cloud requests while maintaining image quality. The framework is supported by an interval-enhanced decoding module and a distributionally robust training strategy to align device and cloud models. Experiments show that CIAR achieves up to 2.18× speed-up and reduces cloud requests by 70% compared to state-of-the-art baselines.

**Strengths:**

1. Practical Impact: The method effectively addresses key bottlenecks in on-device AR image generation (network latency, computational cost) and offers a deployable solution.

2. Comprehensive Evaluation: Extensive experiments across multiple models (LlamaGen, Anole) and metrics (FID, CLIP, speed-up) convincingly demonstrate the advantages over strong baselines.

3. Theoretical Grounding: The appendix provides rigorous mathematical justification for the uncertainty metric and interval fusion operator.

**Weaknesses:**

1. Ablation Clarity: While ablation studies are included, the relative contribution of each component (Inter-Head, prefix injection, Inter-DRO loss) to the overall performance could be more clearly disentangled.

2. Generalization: Experiments are limited to specific AR architectures (LlamaGen, Anole); it's unclear how well CIAR generalizes to other AR or non-AR generative models.

**Questions:**

N/A

---

> ### Author Response · Authors · 2025-12-03
> **Reply to Reviewer QsGz (1/2)**
>
> Dear Reviewer QsGz,
>
> We sincerely thank you for recognizing the potential of our work. We are encouraged by your interest in the detailed mechanisms and the generalization capabilities of CIAR. We have provided a detailed disentanglement of our ablation studies and expanded our experiments to other architectures below.
>
> ---
>
> > **Q1:** Ablation Clarity: While ablation studies are included, the relative contribution of each component (Inter-Head, prefix injection, Inter-DRO loss) to the overall performance could be more clearly disentangled.
>
> **A1:** We appreciate your attention to the internal components of CIAR. Below, we disentangle the contribution and sensitivity of each module.
>
> **1. Inter-Head Capacity:** We investigated how the capacity of the Inter-Head affects the trade-off between latency and accuracy using both **Ensemble** [1] and **Voting** [2] strategies across 1 to 5 heads.
>
> - **Results of Ensemble:**
>
>   | #Heads | CLIP (↑) | FID (↓) | F1(↑)  | HPSv2(↑) | Latency(s) | Steps | Cloud Call |
>   | :------ | :------- | :------ | :----- | :------- | :--------- | :---- | :--------- |
>   | 1 Head  | 0.3159   | 24.2459 | 0.5997 | 22.48    | x2.53      | x3.00 | 30.44%     |
>   | 2 Heads | 0.3157   | 24.1517 | 0.5878 | 22.38    | x2.46      | x2.98 | 29.45%     |
>   | 3 Heads | 0.3151   | 24.1301 | 0.6099 | 22.30    | x2.45      | x2.97 | 28.86%     |
>   | 4 Heads | 0.3137   | 24.6195 | 0.5999 | 22.26    | x2.49      | x2.97 | 28.73%     |
>   | 5 Heads | 0.3147   | 24.6723 | 0.5988 | 22.26    | x2.50      | x3.00 | 28.38%     |
>
> * **Results of Voting:**
>
>   | #Heads | CLIP (↑) | FID (↓) | F1(↑)  | HPSv2(↑) | Latency(s) | Steps | Cloud Call |
>   | :------ | :------- | :------ | :----- | :------- | :--------- | :---- | :--------- |
>   | 1 Head  | 0.3159   | 24.2459 | 0.5997 | 22.48    | x2.53      | x3.00 | 30.44%     |
>   | 2 Heads | 0.3167   | 24.0400 | 0.6095 | 22.34    | x2.29      | x2.98 | 29.38%     |
>   | 3 Heads | 0.3165   | 24.1466 | 0.6195 | 22.25    | x2.48      | x3.00 | 28.97%     |
>   | 4 Heads | 0.3169   | 24.3542 | 0.5678 | 22.25    | x2.53      | x2.98 | 28.56%     |
>   | 5 Heads | 0.3135   | 24.3353 | 0.5910 | 22.22    | x2.48      | x2.98 | 28.54%     |
>
> * **Analysis:**
>   * **Latency Cost:** The latency overhead from adding heads is marginal due to their lightweight parameterization compared to the intermediate layers.
>   * **Performance Trend:** As shown in the tables below, performance peaks at 1-3 heads. However, scaling beyond this (4-5 heads) leads to performance deterioration across CLIP, FID, and HPSv2. This indicates that excessive capacity in the uncertainty estimator does not yield better guidance and may lead to overfitting.
>
> 2. **Analysis of Prefix Injection:** We further analyze the dynamics shown in **Figure 3** regarding two key aspects:
>   - **Trade-off in Speed:** The total inference speed depends on finding a balance between the cloud and the device. At a prefix rate of 0.6, we reach an **optimal balance**. Here, the prefix provides enough context for the device model to generate tokens very quickly. This acceleration on the device side fully compensates for the extra time the cloud spent generating the prefix. However, if the prefix is too long (rate > 0.6), the cloud takes too much time, and the device's speedup cannot offset this cost, causing the overall speed to drop.
>   - **Reduction in Cloud Calls:** The number of cloud requests decreases consistently as the prefix rate increases. This is straightforward: a longer prefix provides richer semantic information. With better context, the device model becomes more confident and accurate, significantly reducing the need to ask the cloud for further corrections.
>
> **3. Inter-DRO Loss: Functional Disentanglement:** To clarify the distinct role of each loss component, we performed an ablation study (results in the table below).
>
> | Method | CLIP (↑) | FID (↓) | F1 (↑) | HPSv2 (↑) |
> | :--- | :--- | :--- | :--- | :--- |
> | w/o $L_c$ | 0.3149 | 25.2721 | 0.5889 | 21.92 |
> | w/o $L_u$ | 0.2891 | 47.8981 | 0.4110 | 16.65 |
> | w/o $L_l$ | 0.2945 | 25.2849 | 0.5889 | 22.45 |
> | w/o $L_\text{anchor}$ | 0.2168 | 68.6400 | 0.4114 | 15.78 |
> | **Ours** | **0.3159** | **24.2459** | **0.5997** | **22.48** |
>
> * **$L_\text{anchor}$ (Basic Convergence):** This ensures the fundamental classification accuracy of the Inter-Head. Removing it prevents the model from converging, leading to a catastrophic drop in image quality (FID 68.64).
> * **$L_c$ (Distribution Alignment):** $L_c$ uses KL-divergence to align the probability distributions between the device and cloud models, ensuring semantic consistency.
> * **$L_l$ (Uncertainty Capture):** This term encourages the Inter-Head to identify uncertain tokens. Its absence results in a slight performance decrease.
> * **$L_u$ (Target Selection):** This governs the precise selection of the correct tokens. Removing it significantly impairs the collaborative mechanism, causing a sharp decline in performance (FID 47.90).
>
> ---

---

> ### Author Response · Authors · 2025-12-03
> **Reply to Reviewer QsGz (2/2)**
>
> > **Q2:** Generalization: Experiments are limited to specific AR architectures (LlamaGen, Anole); it's unclear how well CIAR generalizes to other AR or non-AR generative models.
>
> **A2:** We appreciate this opportunity to demonstrate the generality of CIAR. We extended our framework to a  Next-Scale-Prediction paradigm AR model **(VAR)** [3] and a diffusion-based model **(DiT)** [4].
>
> - **Experiment Setup:** We evaluated performance on ImageNet-1K (50K samples), reporting IS, FID, and Latency.
> - **Results & Analysis:**
>
>   - **Experiments on VAR:** CIAR effectively accelerates the VAR model with minimal quality loss. However, since the Next-Scale Prediction (NSP) paradigm involves shorter sequences, the frequent cloud verification requests in CIAR result in slightly lower acceleration compared to CoDe [5].
>
>     | Method   | IS (↑) | FID (↓) | Precision (↑) | Recall (↑) | Latency      |
>     | :------- | :----- | :------ | :------------ | :--------- | :----------- |
>     | VAR-d24  | 311    | 2.11    | 0.82          | 0.59       | x1.00        |
>     | VAR-d20  | 301    | 2.61    | 0.83          | 0.56       | x1.65        |
>     | CoDe     | 297    | 2.26    | 0.80          | 0.60       | **x2.90**    |
>     | **Ours** | 290    | 2.34    | 0.81          | 0.60       | x2.60 |
>
>   - **Experiments on DiT:** For diffusion models, we adapted the device model to a single DiT block with an Inter-Head to predict noise and variance for the next timestep. CIAR achieves significant acceleration: it matches the image quality of **DDIM-50** (FID 2.32 vs 2.27) yet runs **4.9x faster**.
>
>     | Method        | IS (↑)  | FID (↓)  | Latency   |
>     | :------------ | :------ | :------- | :-------- |
>     | DDIM-50 steps | 241     | 2.27     | x1.00     |
>     | DDIM-25 steps | 232     | 3.18     | x2.00     |
>     | DDIM-20 steps | 221     | 3.81     | x2.50     |
>     | DDIM-12 steps | 184     | 7.80     | x4.17     |
>     | **Ours**      | 240 | 2.32 | **x4.90** |
>
> **Reference:**
>
> [1] Efficient Process Reward Model Training via Active Learning. https://arxiv.org/abs/2504.10559
>
> [2] ANDHRA Bandersnatch: Training Neural Networks to Predict Parallel Realities. https://arxiv.org/abs/2411.19213
>
> [3] Visual Autoregressive Modeling: Scalable Image Generation via Next-Scale Prediction. https://arxiv.org/abs/2404.02905
>
> [4] Scalable Diffusion Models with Transformers. https://arxiv.org/abs/2212.09748
>
> [5] Collaborative Decoding Makes Visual Auto-Regressive Modeling Efficient. https://arxiv.org/abs/2411.17787
>
> ---
>
> Once again, we are truly grateful for your encouraging feedback. We believe the revisions and added analyses have strengthened the manuscript, and we hope you find our response satisfactory.
>
> Sincerely,
>
> The Authors

---

### Official Review · Reviewer_t33Y · 2025-10-30

**Soundness:** 2
**Presentation:** 3
**Contribution:** 3
**Rating:** 4
**Confidence:** 4

**Summary:**

This paper proposes CIAR (Collaborative Interval-based AutoRegressive Decoding), a cloud-based collaborative framework for accelerating autoregressive image generation. Traditional visual autoregressive models are slow and difficult to deploy on devices. CIAR introduces an interval-based uncertainty quantization head on the device, replacing discrete predictions with continuous probability intervals, thereby efficiently determining which tokens are acceptable locally and which need to be uploaded to the cloud for verification. Experimental results show that CIAR achieves a significant acceleration and reduction in cloud calls while maintaining image quality, improving the practicality and efficiency of visual autoregressive image generation.

**Strengths:**

1. This article is written clearly and logically.

2. This article analyzes the challenges of deploying autoregressive image generation models on devices and proposes an edge-cloud collaborative framework.

3. The proposed uncertainty estimation method is interesting.

4. Comprehensive experiments demonstrate the superiority of the proposed method compared to other acceleration baselines and its advantage over other uncertainty estimation methods.

**Weaknesses:**

1. The paper lacks details on efficiency measurement. What hardware was used for latency and speedup testing? This seems to be unmentioned in the paper.

2. As an edge-cloud collaborative method, what is the communication cost of CIAR? Are there any latency tests conducted by deploying CIAR in a real edge-cloud collaborative scenario?

3. Can the proposed method also be applied to the VAR model of next-scale prediction?

**Questions:**

Please see the weakness

---

> ### Author Response · Authors · 2025-12-03
> **Reply to Reviewer t33Y (1/2)**
>
> Dear Reviewer t33Y,
>
> We sincerely thank you for your thoughtful review. We have addressed your questions regarding the hardware specifications, device-cloud communication costs, and the scalability of CIAR below.
>
> ---
>
> > **Q1:** The paper lacks details on efficiency measurement. What hardware was used for latency and speedup testing? This seems to be unmentioned in the paper.
>
> **A1:** We apologize for the omission. All model training, inference latency tests, and speedup evaluations were conducted on **NVIDIA GeForce RTX 4090 GPU (24GB VRAM)**. We have added this specification to the experimental setup section in the revised manuscript.
>
> ---
>
> > **Q2:** As an edge-cloud collaborative method, what is the communication cost of CIAR? Are there any latency tests conducted by deploying CIAR in a real edge-cloud collaborative scenario?
>
> **A2:** Due to resource constraints preventing a physical large-scale deployment, we followed established protocols from classic cloud-device collaboration works [1, 2, 3, 4] to simulate the communication environment.
>
> **1. Simulation Setup:** We evaluated performance under three representative network profiles (5G, 4G, WiFi). The communication cost and total latency are calculated using the following formulations:
>
> $$
> T_{\text{comm}} = L + \frac{\text{DataSize}}{B}
> $$
>
> $$
> T_{\text{total}} = T_{\text{comm}}^{\text{cloud}} + T_{\text{comm}}^{\text{device}} + T_{\text{cloud}} + T_{\text{device}}
> $$
>
> where $L$ denotes the network latency, $B$ represents the bandwidth, and $\text{DataSize}$ refers to the transmission payload size. Additionally, $T_{\text{cloud}}$ and $T_{\text{device}}$ represent the computational latency on the cloud and device, respectively. The specific network configurations are:
> * **5G:** Bandwidth = 300.00 Mbps, Latency = 10.00 ms
> * **4G:** Bandwidth = 20.00 Mbps, Latency = 50.00 ms
> * **WiFi:** Bandwidth = 100.00 Mbps, Latency = 20.00 ms
>
> **2. Results & Analysis:** We compared CIAR with the baseline method, Lantern. As shown in the table below, CIAR significantly reduces the communication ratio and total latency across all network conditions.
>
> * **Reduced Overhead:** By accurately identifying certain tokens, CIAR minimizes unnecessary data transmission, resulting in a significantly lower communication ratio (Comm/Tot) compared to Lantern (e.g., **0.247 vs. 0.276** under 5G).
> * **Cloud Offloading:** Our method effectively reduces the computational load on the cloud, leading to faster total inference.
>
> | Network  | Method   | Device (ms) | Cloud (ms) | Communication (ms) | Total (ms)   | Ratio (Comm/Tot) |
> | :------- | :------- | :---------- | :--------- | :----------------- | :----------- | :--------------- |
> | **5G**   | Lantern  | 11.63       | 11586.29   | 4424.63            | 16022.54     | 0.2762           |
> |          | **Ours** | 13.54       | 5102.60    | **1678.13**        | **6794.27**  | **0.2470**       |
> | **4G**   | Lantern  | 11.90       | 11577.91   | 22291.31           | 33881.12     | 0.6579           |
> |          | **Ours** | 14.10       | 4847.20    | **8173.95**        | **13035.26** | **0.6271**       |
> | **WiFi** | Lantern  | 11.69       | 11436.11   | 8768.01            | 20215.81     | 0.4337           |
> |          | **Ours** | 14.11       | 5052.52    | **3343.94**        | **8410.58**  | **0.3976**       |
>
> ---

---

> ### Author Response · Authors · 2025-12-03
> **Reply to Reviewer t33Y (2/2)**
>
> > **Q3:** Can the proposed method also be applied to the VAR model of next-scale prediction?
>
> **A3:** Yes, CIAR is a general framework applicable to the VAR (Next-Scale Prediction) paradigm.
>
> **1. Implementation on VAR:** We adapted CIAR to VAR by utilizing pre-trained models of different scales as the device and cloud models, specifically fine-tuning the prediction head for the final scale features.
>
> **2. Results & Discussion:** The comparison results are presented below:
>
> | Method   | IS (↑) | FID (↓) | Precision (↑) | Recall (↑) | Latency      |
> | :------- | :----- | :------ | :------------ | :--------- | :----------- |
> | VAR-d24  | 311    | 2.11    | 0.82          | 0.59       | x1.00        |
> | VAR-d20  | 301    | 2.61    | 0.83          | 0.56       | x1.65        |
> | CoDe     | 297    | 2.26    | 0.80          | 0.60       | **x2.90**    |
> | **Ours** | 290    | 2.34    | 0.81          | 0.60       | x2.60 |
>
> * **Effectiveness:** CIAR successfully accelerates the VAR model (**x2.60**) while maintaining competitive generation quality (FID 2.34), demonstrating the extensibility of our method.
> * **Comparison with CoDe:** We observe that CoDe achieves slightly better performance in this specific setting. This is expected as CoDe is natively designed for the **Next-Scale-Prediction** paradigm. The sequence length in VAR is relatively short (scales), meaning frequent cloud requests in CIAR can introduce overhead that offsets some acceleration gains compared to CoDe's parallel decoding approach. However, CIAR remains a robust alternative with strong performance, particularly for autoregressive tasks.
>
> **Reference:**
>
> [1] Neurosurgeon: Collaborative Intelligence Between the Cloud and Mobile Edge. https://dl.acm.org/doi/abs/10.1145/3093337.3037698
>
> [2] SPINN: Synergistic Progressive Inference of Neural Networks over Device and Cloud. https://arxiv.org/pdf/2008.06402
>
> [3] JALAD: Joint Accuracy- and Latency-Aware Deep Structure Decoupling for Edge-Cloud Execution. https://arxiv.org/pdf/1812.10027
>
> [4] DUET: A Tuning-Free Device-Cloud Collaborative Parameters Generation Framework for Efficient Device Model Generalization. https://arxiv.org/pdf/2209.05227
>
> ---
>
> We sincerely appreciate your insightful comments, which have significantly helped us improve the quality and clarity of our work. We hope our revisions meet your expectations.
>
> Sincerely,
>
> The Authors

---

### Official Review · Reviewer_vW4B · 2025-11-06

**Soundness:** 2
**Presentation:** 2
**Contribution:** 2
**Rating:** 4
**Confidence:** 5

**Summary:**

The paper proposes CIAR, a cloud-device collaborative interval-based decoding framework designed to accelerate image generation with auto-regressive (AR) models. The method introduces an interval-based uncertainty quantifier (“Inter-Head”) for the device model to filter highly confident tokens, reducing the need for cloud verification. CIAR incorporates a continuous interval-based uncertainty measure, cloud-enhanced decoding with interval-feature conditioning, and an interval-aware DRO loss for distribution alignment between device and cloud outputs. Experimental results on text-to-image tasks demonstrate that CIAR substantially reduces latency and the number of cloud requests while maintaining comparable image quality to state-of-the-art speculative decoding baselines.

**Strengths:**

The paper clearly identifies bottlenecks in AR-based image generation for on-device deployment, specifically the challenge of excessive cloud requests and inefficiency of uniform token verification, and motivates a focused solution.

The introduction of interval-based uncertainty (via the “Inter-Head”) in the device model is a creative step toward efficiently identifying tokens that can be safely accepted on-device, addressing the challenge posed by large visual vocabularies and spatial redundancy.

The paper goes a long way in justifying and formalizing the proposed continuous interval-based uncertainty metric, providing detailed theoretical rationale, axiomatic desiderata, and mathematical properties (see Appendix A.3). This is elaborated with specific attention to the geometric/statistical meaning of the measure, which demonstrates a serious attempt at rigor (even if some aspects could benefit from deeper empirical probing).

**Weaknesses:**

Despite the strong baseline coverage, the paper omits a direct comparison and discussion of several directly relevant. In particular, work such as “Collaborative Decoding Makes Visual Auto-Regressive Modeling Efficient” (Chen et al., 2025) is highly relevant and should be both cited and empirically compared (e.g., as a baseline in Tables 1 and 2, and in distributional alignment discussions). The failure to engage with such work weakens the claim of CIAR’s advancement.

While the role of prefix rate and threshold is explored (Figure 3, Figure 6), there is little sensitivity analysis on device model depth, Inter-Head capacity, or how interval regularization influences the tradeoff between speed and quality. For instance, it is unclear how the single-layer device model limits or shapes the uncertainty estimation capabilities—could more layers or changes to Inter-Head’s complexity yield diminishing returns or instability?

While the Inter-DRO loss is well-motivated mathematically, the practical effect of its components is not independently quantified nor are variants (e.g., anchor loss alone vs. full Inter-DRO) ablated. Table 1 does not include or discuss what is lost if alignment is not enforced (e.g., mode collapse, sample drift?), and Figure 4 only vaguely notes “consistency” improvements. A major claim—alignment between cloud and device distributions—lacks direct quantitative substantiation or specific examples measuring divergences before/after.

**Questions:**

See weaknesses.

---

> ### Author Response · Authors · 2025-12-03
> **Reply to Reviewer vW4B (1/3)**
>
> Dear Reviewer vW4B,
>
> We sincerely thank you for your constructive suggestions regarding the comparison with related work and the ablation analysis. We have carefully addressed your concerns and added the requested experiments below.
>
> ---
>
> > **Q1:** Despite the strong baseline coverage, the paper omits a direct comparison and discussion of several directly relevant. In particular, work such as “Collaborative Decoding Makes Visual Auto-Regressive Modeling Efficient” (Chen et al., 2025) is highly relevant and should be both cited and empirically compared (e.g., as a baseline in Tables 1 and 2, and in distributional alignment discussions). The failure to engage with such work weakens the claim of CIAR’s advancement.
>
> **A1:** We acknowledge the significance of CoDe [1] in collaborative acceleration. We have cited this work and added it as a baseline in the revised **Table 1**. Below, we provide a detailed comparison and analysis.
>
> **1. Comparison between CoDe and CIAR**
>
> * **Experimental Setup:** It is important to note that CoDe targets the **Next-Scale-Prediction (VAR)** [2] paradigm, which differs fundamentally from the **Next-Token-Prediction (LlamaGen)** paradigm used in our work. To ensure a fair comparison, we adapted CoDe to the LlamaGen framework, where the parameter $N$ represents the ratio of tokens generated by the cloud model.
>
> * **Results:** The comparison is shown below:
>
>   | Methods        | CLIP (↑) | FID (↓) | F1 (↑) | HPSv2 (↑) | Latency   | Steps     | Cloud Call |
>   | :------------- | :------- | :------ | :----- | :-------- | :-------- | :-------- | :--------- |
>   | Base           | 0.3161   | 23.6900 | 0.6097 | 22.74     | x1.00     | x1.00     | 100.00%    |
>   | CoDe ($N=0.3$) | 0.2827   | 35.6670 | 0.4625 | 18.08     | x2.04     | x2.734    | 30.00%     |
>   | **Ours**       | 0.3159   | 24.2459 | 0.5997 | 22.48     | **x2.53** | **x3.00** | 30.44%     |
>
> * **Analysis:** While CoDe accelerates generation, its performance drops significantly in the Next-Token-Prediction paradigm. In VAR (Next-Scale), the initial scales represent global image semantics; the cloud model establishes a strong semantic foundation. However, in LlamaGen (NTP), tokens are generated sequentially. If the cloud only provides the first $N$ tokens, the small device model lacks sufficient context to sustain long-term consistency, leading to severe **distribution shift** in later tokens. CIAR addresses this by dynamically intervening based on uncertainty.
>
> **2. Analysis of Different Cloud Ratios ($N$)** : We further analyzed the impact of varying $N$ in the CoDe adaptation. Below is the result:
>
> | Methods         | CLIP (↑) | FID (↓) | F1 (↑) | HPSv2 (↑) | Latency   | Cloud Call |
> | :-------------- | :------- | :------ | :----- | :-------- | :-------- | :--------- |
> | Base            | 0.3161   | 23.6900 | 0.6097 | 22.74     | x1.00     | 100.00%    |
> | CoDe ($N=0.10$) | 0.2566   | 44.7305 | 0.4028 | 15.84     | x2.71     | 10.00%     |
> | CoDe ($N=0.15$) | 0.2620   | 42.6082 | 0.4088 | 16.36     | x2.41     | 15.00%     |
> | CoDe ($N=0.20$) | 0.2627   | 40.3943 | 0.4508 | 16.83     | x2.14     | 20.00%     |
> | CoDe ($N=0.25$) | 0.2783   | 38.3103 | 0.4639 | 17.43     | x2.13     | 25.00%     |
> | CoDe ($N=0.30$) | 0.2827   | 35.6670 | 0.4625 | 18.08     | x2.04     | 30.00%     |
> | **Ours**        | 0.3159   | 24.2459 | 0.5997 | 22.48     | **x2.53** | 30.44%     |
>
> Increasing $N$ (more cloud involvement) improves image quality but significantly diminishes the acceleration gain (Latency drops from x2.71 to x2.04). CIAR performs better between quality and speed.
>
> ---

---

> ### Author Response · Authors · 2025-12-03
> **Reply to Reviewer vW4B (2/3)**
>
> > **Q2:** While the role of prefix rate and threshold is explored (Figure 3, Figure 6), there is little sensitivity analysis on device model depth, Inter-Head capacity, or how interval regularization influences the tradeoff between speed and quality. For instance, it is unclear how the single-layer device model limits or shapes the uncertainty estimation capabilities—could more layers or changes to Inter-Head’s complexity yield diminishing returns or instability?
>
> **A2:** We appreciate this insightful suggestion regarding model sensitivity. We have conducted ablation studies on both device model depth and Inter-Head capacity.
>
> **1. Analysis of Device Model Depth:** We trained device models with 1, 2, 3, and 5 layers under identical configurations.
>
> | #Layers | CLIP (↑) | FID (↓) | F1 (↑) | HPSv2 (↑) | Latency | Steps | Cloud Call |
> | :--- | :--- | :--- | :--- | :--- | :--- | :--- | :--- |
> | 1 Layer | 0.3159 | 24.2459 | 0.5997 | 22.48 | x2.53 | x3.00 | 30.44% |
> | 2 Layers | 0.3113 | 23.8820 | 0.6270 | 22.44 | x2.54 | x3.44 | 27.06% |
> | 3 Layers | 0.3171 | 23.8404 | 0.5899 | 22.44 | x2.19 | x3.58 | 26.10% |
> | 5 Layers | 0.3171 | 23.8232 | 0.6180 | 22.45 | x2.14 | x3.66 | 25.68% |
>
> **Analysis:** Increasing the device model depth improves image quality and reduces the frequency of cloud calls due to enhanced local capabilities. However, this accuracy gain comes at the cost of efficiency, as the added computational overhead on the device increases overall latency.
>
> **2. Analysis of Inter-Head Capacity:** We investigated the impact of Inter-Head capacity using two strategies: **Ensemble** [3] and **Voting** [4], across 1 to 5 heads initialized with different seeds.
>
> - **Results of Ensemble:**
>
>   | #Heads | CLIP (↑) | FID (↓) | F1(↑)  | HPSv2(↑) | Latency(s) | Steps | Cloud Call |
>   | :------ | :------- | :------ | :----- | :------- | :--------- | :---- | :--------- |
>   | 1 Head  | 0.3159   | 24.2459 | 0.5997 | 22.48    | x2.53      | x3.00 | 30.44%     |
>   | 2 Heads | 0.3157   | 24.1517 | 0.5878 | 22.38    | x2.46      | x2.98 | 29.45%     |
>   | 3 Heads | 0.3151   | 24.1301 | 0.6099 | 22.30    | x2.45      | x2.97 | 28.86%     |
>   | 4 Heads | 0.3137   | 24.6195 | 0.5999 | 22.26    | x2.49      | x2.97 | 28.73%     |
>   | 5 Heads | 0.3147   | 24.6723 | 0.5988 | 22.26    | x2.50      | x3.00 | 28.38%     |
>
> * **Results of Voting:**
>
>   | #Heads | CLIP (↑) | FID (↓) | F1(↑)  | HPSv2(↑) | Latency(s) | Steps | Cloud Call |
>   | :------ | :------- | :------ | :----- | :------- | :--------- | :---- | :--------- |
>   | 1 Head  | 0.3159   | 24.2459 | 0.5997 | 22.48    | x2.53      | x3.00 | 30.44%     |
>   | 2 Heads | 0.3167   | 24.0400 | 0.6095 | 22.34    | x2.29      | x2.98 | 29.38%     |
>   | 3 Heads | 0.3165   | 24.1466 | 0.6195 | 22.25    | x2.48      | x3.00 | 28.97%     |
>   | 4 Heads | 0.3169   | 24.3542 | 0.5678 | 22.25    | x2.53      | x2.98 | 28.56%     |
>   | 5 Heads | 0.3135   | 24.3353 | 0.5910 | 22.22    | x2.48      | x2.98 | 28.54%     |
>
> - **Analysis:**
>   - **Computation Cost:** Increasing the number of heads has a negligible impact on latency due to their small parameter size relative to the model layers.
>
>   - **Diminishing Returns:** While FID declines slightly with 2-3 heads, performance degrades (across CLIP, FID, HPSv2) when scaling to 5 heads. This suggests that excessive Inter-Head capacity does not yield better uncertainty estimation and may lead to overfitting or instability.
>
> ---

---

> ### Author Response · Authors · 2025-12-03
> **Reply to Reviewer vW4B (3/3)**
>
> > **Q3:** While the Inter-DRO loss is well-motivated mathematically, the practical effect of its components is not independently quantified nor are variants (e.g., anchor loss alone vs. full Inter-DRO) ablated. Table 1 does not include or discuss what is lost if alignment is not enforced (e.g., mode collapse, sample drift?), and Figure 4 only vaguely notes “consistency” improvements. A major claim—alignment between cloud and device distributions—lacks direct quantitative substantiation or specific examples measuring divergences before/after.
>
> **A3:** This is a very constructive suggestion. To quantify the contribution of each component in the **Inter-DRO loss**, we performed a comprehensive ablation study. The results are summarized below:
>
> | Method           | CLIP (↑)   | FID (↓)     | F1 (↑)     | HPSv2 (↑) |
> | :--------------- | :--------- | :---------- | :--------- | :-------- |
> | w/o $L_c$        | 0.3149     | 25.2721     | 0.5889     | 21.92     |
> | w/o $L_u$        | 0.2891     | 47.8981     | 0.4110     | 16.65     |
> | w/o $L_l$        | 0.2945     | 25.2849     | 0.5889     | 22.45     |
> | w/o $L_\text{anchor}$ | 0.2168     | 68.6400     | 0.4114     | 15.78     |
> | **Ours**         | **0.3159** | **24.2459** | **0.5997** | **22.48** |
>
> **Analysis of Loss Components:**
> 1.  **$L_\text{anchor}$ (Crucial for Convergence):** This loss ensures the basic classification accuracy of the Inter-Head. Removing it causes a failure in convergence and a catastrophic drop in image quality (FID spikes to 68.64).
> 2.  **$L_c$ (Distribution Alignment):** Unlike $L_\text{anchor}$ which targets head accuracy, $L_c$ utilizes KL-divergence to explicitly align the probability distributions between the device and cloud models. Removing it leads to a noticeable drop in consistency (FID increases by ~1.0).
> 3.  **$L_l$ (Uncertainty Capture):** This component aids the Inter-Head in identifying uncertain tokens. Removing it results in a slight performance degradation.
> 4.  **$L_u$ (Target Selection):** This loss governs the accurate selection of target tokens. Its removal significantly impacts the generation quality (FID increases to 47.89), proving its vital role in the collaborative mechanism.
>
> **Reference:**
>
> [1] Collaborative Decoding Makes Visual Auto-Regressive Modeling Efficient. https://arxiv.org/abs/2411.17787
>
> [2] Visual Autoregressive Modeling: Scalable Image Generation via Next-Scale Prediction. https://arxiv.org/abs/2404.02905
>
> [3] Efficient Process Reward Model Training via Active Learning. https://arxiv.org/abs/2504.10559
>
> [4] ANDHRA Bandersnatch: Training Neural Networks to Predict Parallel Realities. https://arxiv.org/abs/2411.19213
>
> ---
>
> We sincerely hope that our response and the additional experiments have adequately addressed your concerns.
>
> Sincerely,
>
> The Authors

---

### Author Response · Authors · 2025-12-03
**Reply to All Reviewers**

Dear Reviewers,

We sincerely thank all reviewers for their insightful feedback and constructive suggestions. We are highly encouraged that the reviewers recognized the **creativity** of CIAR (Reviewer vW4B), acknowledged its **superiority** over existing acceleration methods (Reviewer t33Y), and highlighted its **practical impact** (Reviewer QsGz and Reviewer g7v5). We are also grateful for the positive remarks regarding our **theoretical grounding** for the uncertainty metric (Reviewer vW4B and Reviewer QsGz) and the **clarity of our writing** (Reviewer t33Y).

In this rebuttal, we have addressed all raised concerns through the following improvements:

* **Comprehensive Baseline Comparison:** We have incorporated the state-of-the-art **CoDe** method as a strong baseline to rigorously benchmark our performance. Specifically, we conducted a comparative analysis between CoDe and CIAR under various parameter configurations, demonstrating the superiority of our approach. These results have been added to **Table 1** and **Section A.2.3** of the revised manuscript.
* **In-depth Ablation Analysis:** We provided extensive ablation studies covering model architecture (depth, capacity) and training strategies (loss components) to validate the effectiveness of each module. Detailed results are provided in **Sections A.2.4, A.2.5, and A.2.6**.
* **Rigorous Communication Simulation:** We conducted quantitative simulations of **edge-cloud communication costs** under 5G, 4G, and WiFi environments. These tests verify the efficiency and practicality of CIAR in real-world deployments, proving that our method significantly alleviates the cloud-side computational load. The corresponding experiments are detailed in **Section A.2.7**.
* **Quantified Generalization:** We successfully extended CIAR to the **Next-Scale Prediction (VAR)** paradigm and **Diffusion Models (DiT)**. CIAR exhibits strong collaborative acceleration capabilities on both architectures, demonstrating significant generalization across diverse generative models. These experiments are included in **Section A.2.8**.

We hope our responses and additional experiments satisfactorily address your questions.

Sincerely,
The Authors

---

### Author Response · Authors · 2025-12-03
**Summary of Rebuttal Updates**

Dear Reviewers, AC, SAC, and PC,

We sincerely thank you for the time and effort dedicated to reviewing our paper. We are encouraged for  the reviewers' recognition of CIAR's **creativity** (Reviewer vW4B), **superior efficiency** (Reviewer t33Y), **practical impact** (Reviewer QsGz, g7v5), **theoretical grounding** (Reviewer vW4B, QsGz), and **writing clarity** (Reviewer t33Y).

> **Summary of Contributions:** CIAR leverages an cloud-device collaborative architecture and employs the proposed **Inter-DRO loss** to train an **Inter-Head**. This mechanism enables the calculation of probability intervals for precise uncertainty quantification. Compared to baselines, CIAR significantly accelerates generation without compromising image quality. Furthermore, the CIAR framework demonstrates strong extensibility to **Next-Token-Prediction (NTP)** autoregressive architectures, **Next-Scale-Prediction (NSP)** autoregressive architectures, and **Diffusion-based** architectures.

---

### **1. Superior Acceleration and Reliable Uncertainty Quantification**

Compared to standard large model generation, our CIAR method achieves a **2.53x speedup** without significant degradation in image quality. Crucially, it reduces cloud calls to just **30.44%**, significantly alleviating the computational load on the cloud side.

| Methods | CLIP (↑) | FID (↓) | F1 (↑) | HPSv2 (↑) | Latency | Steps | Cloud Call |
| :--- | :--- | :--- | :--- | :--- | :--- | :--- | :--- |
| Base | 0.3161 | 23.6900 | 0.6097 | 22.74 | x1.00 | x1.00 | 100.00% |
| CoDe ($N=0.3$) | 0.2827 | 35.6670 | 0.5825 | 18.08 | x2.04 | x2.734 | 30.00% |
| **Ours** | 0.3159 | 24.2459 | 0.5997 | 22.48 | **x2.53** | **x3.00** | 30.44% |

Our proposed **Interval-based** uncertainty quantification method, which utilizes output probability intervals, demonstrates superior performance compared to existing uncertainty estimation techniques.

| Methods | CLIP (↑) | FID (↓) | F1(↑) | HPSv2(↑) | Speedup | Steps | Cloud Call |
| :--- | :---: | :---: | :---: | :---: | :---: | :---: | :---: |
| Base | 0.3161 | 23.6900 | 0.6097 | 22.74 | x1.00 | x1.00 | 100.00% |
| Entropy-Lens | 0.3132 | 24.5828 | 0.5796 | 22.03 | x1.70 | x2.05 | 52.34% |
| SoftmaxCorr | 0.3149 | 31.1009 | 0.5130 | 19.11 | x2.27 | x2.31 | 36.49% |
| **Ours** | 0.3159 | 24.2459 | 0.5997 | 22.48 | **x2.53** | **x3.00** | **30.44%** |

---

### **2. Remarkable Generalization**

CIAR excels not only in **Next-Token-Prediction (NTP)** autoregressive architectures but also extends effectively to **Next-Scale-Prediction (NSP)** autoregressive models and **Diffusion-based** models. This confirms CIAR as a generalizable framework for accelerating image generation.
| Method | IS (↑) | FID (↓) | Precision (↑) | Recall (↑) | Latency |
| :--- | :---: | :---: | :---: | :---: | :---: |
| VAR-d24 | 311 | 2.11 | 0.82 | 0.59 | x1.00 |
| VAR-d20 | 301 | 2.61 | 0.83 | 0.56 | x1.65 |
| **Ours** | 290 | 2.34 | 0.81 | 0.60 | x2.60 |

| Method | IS (↑) | FID (↓) | Latency |
| :--- | :---: | :---: | :---: |
| DDIM-50 steps | 241 | 2.27 | x1.00 |
| DDIM-25 steps | 232 | 3.18 | x2.00 |
| **Ours** | 240 | 2.32 | **x4.90** |

---

### **3. Key Improvements in Rebuttal**
In this rebuttal, we have carefully addressed all concerns optimized the manuscript with the following key updates:

* ***Incorporated SOTA Baseline:*** Added a rigorous comparison with **CoDe**, analyzing performance across various parameter configurations to validate CIAR's superiority (**Table 1, Sec A.2.3**).
* ***Deepened Ablation Studies:*** Expanded experiments on **model architecture** (depth, capacity) and **training strategies** (loss components) to isolate the effectiveness of each module (**Sec A.2.4 - A.2.6**).
* ***Simulated Real-world Deployment:*** Conducted quantitative simulations of **Cloud-Device communication** under 5G, 4G, and WiFi settings, verifying practical efficiency and reduced cloud load (**Sec A.2.7**).
* ***Verified Generalization:*** Extended CIAR to **Next-Scale Prediction (VAR)** and **Diffusion Models (DiT)**, demonstrating robust collaborative acceleration across diverse generative architectures (**Sec A.2.8**).

---

We sincerely hope that our supplementary experiments and responses provide a comprehensive understanding of our work. If there are any remaining questions, please feel free to contact us; we are more than happy to address them.

Sincerely,

The Authors

---

### Meta-Review · Area_Chair_NB9E · 2026-01-11

**Summary:**

While most reviewers appreciate the interesting idea of interval-based collaborative decoding, its theoretical grounding, potential practical impact, and good empirical performance, they raised concerns about missing comparisons/analyses and limited evaluation on AR methods. The authors’ rebuttal carefully addresses most of them, providing detailed additional experiments and explanations. For example, in response to reviewer vW4B, the authors provided comparisons with CoDe (Chen et al., 2025), clearly showing the superiority of the proposed method; they also conducted additional ablative studies across depths and capacities and showed the effect of the proposed method on next-scale prediction and diffusion models. AC finds that these results in rebuttal resolve most of the concerns raised by the reviewers, without remaining critical outstanding concerns. Considering the quality of the rebuttal and the shared consensus on the technical contribution of this work, AC recommends acceptance of this paper, encouraging the authors to carefully incorporate the rebuttal contents into the final manuscript.

**Reviewer Concerns:**

The main concerns were missing comparisons/analyses and evaluations limited to AR methods. The rebuttal resolves them by conducting experiments at different depths and capacities in architecture and ablation studies on loss components to isolate the effectiveness of each module. They also conducted quantitative simulations of Cloud-Device communication under 5G, 4G, and WiFi settings, verifying practical efficiency and reduced cloud load. Furthermore, the authors provided additional experiments on Next-Scale Prediction (VAR) and Diffusion Models (DiT), demonstrating robust collaborative acceleration across diverse generative architectures. AC finds that these results in rebuttal resolve most of the concerns raised by the reviewers, without remaining critical outstanding concerns.

**Reviewer Scores:**

Reviewer vW4B  would have changed the score from 4 to 6.
Reviewer t33Y would have changed the score from 4 to 6.
Reviewer vW4B  would have raised the score from 6 to 8.
Reviewer g7v5 would have changed the score from 4 to 6.

---

### Decision · Program_Chairs · 2026-01-26

Accept (Poster)